

# Elastic-viscoplastic characterization of S2 columnar freshwater ice

Iman E. Gharamti[1], John P. Dempsey[2], Arttu Polojärvi[1], and Jukka Tuhkuri[1]

[1]Department of Mechanical Engineering, Aalto University, Espoo 00076, Finland
[2]Department of Civil and Environmental Engineering, Clarkson University, Potsdam, NY 13699, USA

**Correspondence:** Iman E. Gharamti (iman.elgharamti@aalto.fi)

**Abstract.** This work addresses the time-dependent response of 3m x 6m floating edge-cracked rectangular plates of columnar freshwater S2 ice by conducting load control (LC) mode I fracture tests at -2 °C in the Ice Tank of Aalto University. The loading profile consisted of creep/cyclic-recovery sequences followed by a monotonic ramp to fracture. The LC test results were compared with previous monotonically loaded displacement control (DC) experiments of the same ice, and the effect of creep and cyclic sequences on the fracture properties were discussed. To characterize the nonlinear displacement-load relation, Schapery's constitutive model of nonlinear thermodynamics was applied to analyze the experimental data. A numerical optimization procedure using Nelder-Mead's (N-M) method was implemented to evaluate the model functions by matching the displacement record generated by the model and measured by the experiment. The accuracy of the constitutive model is checked and validated against the experimental response at the crack mouth. Under the testing conditions, the creep phases were dominated by a steady phase, and the ice response was elastic-viscoplastic; no viscoelasticity or major recovery were detected. In addition, there was no clear effect of the creep loading on the fracture properties: the apparent fracture toughness, failure load, and crack opening displacements.

## 1 Introduction

The interaction of ice floes with structures is a major challenge for all operations taking place in the Arctic regions. Understanding the mechanics of ice-structure interaction is very important to ensure the safe design and secure operations of ships and offshore structures. This interaction phenomenon involves complicated physical processes and combined deformation modes that should be analyzed in order to answer several engineering questions: such as estimating the ice loads on structures and the necessary forces required for fracturing and navigating through an ice sheet.

Tackling this problem is handicapped by the wide range of time, rate and load scales involved. Ice response is highly dependent on the loading conditions. At high deformation rates, ice behaves in a brittle manner, and fracture controls the loads exerted on structures. At low deformation rates, ice creeps, and the forces are controlled by time-dependent deformation (Palmer et al., 1983; Gharamti et al., in-press). In fact, ice can exhibit a strong time-dependent behavior depending on the operating factors: loading profile, load levels, duration of load/unload applications, temperature, and other environmental factors. Accordingly, precise prediction of ice forces requires an analysis that simultaneously accounts for the time-dependent and fracture behavior of ice.



The time-dependent constitutive modeling has become increasingly important to characterize the mechanical behavior of ice. The time-dependent response of ice is highly nonlinear, and the constitutive model should incorporate elastic, viscoelastic/delayed elastic (time-dependent and recoverable) and viscoplastic/viscous (time-dependent and unrecoverable) components, depending on the engineering problem. For example, glaciologists are interested solely in long term creep behavior, hence an

ice model to them is simply a viscous creep law. Floating ice (freshwater or sea ice) problems generally focus on shorter term behavior with damage/failure so all three components of deformation need to be modeled.

The time-dependent behavior of freshwater ice has been addressed with great attention, and several constitutive models were developed (Michel, 1978; Sinha, 1978; Le Gac and Duval, 1980; Ashby and Duval, 1985; Sunder and Wu, 1989; Mellor and Cole, 1983; Cole, 1990; Duval et al., 1991; Sunder and Wu, 1990; Abdel-Tawab and Rodin, 1997; Santaoja, 1990).

Constitutive laws can be phenomenological or micromechanical. Micromechanical modeling in ice faces challenges because the characterization of the microscopic mechanisms of ice deformation is still inadequate (Abdel-Tawab and Rodin, 1997).

Phenomenological laws are classified into two groups. The first group are empirical-based relations (Sinha, 1978; Schapery, 1969). Their equations relate macroscopic variables: stress/load, strain/displacement, and time. They do not contain state variables that describe the internal state of the material and are valid only for constant stress/load. The functions in these models

can be easily calibrated to simulate the experiments. The second group of phenomenological models starts from physically-based models involving internal state variables (dislocation density, internal stresses reflecting hardening, etc ...); they develop differential equations for the evolution of these variables with time and quantify the dependence of these variables on stress, temperature and strain (Le Gac and Duval, 1980; Sunder and Wu, 1989, 1990; Abdel-Tawab and Rodin, 1997). These models provide insights into the microscopic mechanisms taking place, and the state variables describe the deformation resistance

offered by changes in the microstructure of the material. However, they require a proper identification of the deformation mechanisms.

The effect of time-dependent loading on the strength of freshwater ice has been examined in the literature. Subjecting freshwater ice to cyclic loading apparently leads to a significant increase in the tensile, compressive, and flexural strength of that ice (Murdza et al., 2020; Iliescu et al., 2017; Iliescu and Schulson, 2002; Cole, 1990; Jorgen and Picu, 1998). On the other

hand, no detailed investigation of the effect of creep and cyclic loading on the fracture properties has been conducted in the past.

The goal of this paper is to characterize the time-dependent and fracture behavior of 3m x 6m floating edge-cracked rectangular plates of columnar freshwater S2 ice under creep-recovery loading and monotonic loading to fracture at -2 °C. A program of five load control (LC) mode I fracture tests was completed in the test basin (40 m square and 2.8 m deep) at Aalto university.

Creep and cyclic sequences were applied below the failure loads, followed by monotonic ramps leading to complete fracture of the specimen. The LC results were compared with the fracture results of monotonically loaded displacement control (DC) tests of the same ice (Gharamti et al., in-press), and the effect of the creep and cyclic sequences on the fracture properties were analyzed.

The constitutive modeling used in this paper was presented by Schapery, where it was applied to polymers (Schapery, 1969).

Schapery's model belongs to the first phenomenological group and originates from the theory of nonlinear thermodynamics.





This study presents the first attempt to use Schapery's model for freshwater ice. The choice of this model for freshwater ice is motivated by the fact that the model was successfully applied to saline ice (Schapery, 1997; Adamson and Dempsey, 1998; LeClair et al., 1999, 1996) with encouraging results. The model accurately described the deformation response during load/unload applications over varying load profiles.

The experiments in this study aim to assess the time-dependent nature of warm columnar freshwater S2 ice. Especially, the study aims to examine: 1) the extent to which the elastic, viscoelastic and viscoplastic components contribute to the crack mouth opening displacement, 2) the effects of the testing conditions on the creep stages (primary/transient and steady-state/secondary) present in the ice, 3) the effects that creep and cyclic sequences have on the fracture properties; i.e. apparent fracture toughness, failure load, and crack growth initiation displacements, and 4) the ability of Schapery's nonlinear constitutive model to predict
the experimental response.

The rest of the paper is structured as follows. In Section 2, a description of the experimental setup, testing conditions, and the applied loading profile is presented. Section 3 introduces Schapery's model that is used to analyze the experiments. In section 4, the experimental and model results are summarized and analyzed. Section 5 concludes the paper.

## 2    Creep-recovery fracture experiments

### 2.1    Experimental details

The ice specimens tested were 3m x 6m rectangular plates, cut from a 40m x 40m parent sheet, with a thickness of 340 - 410 mm, and instrumented as shown in Fig. 1. The ice was columnar freshwater S2 ice with an almost linear temperature gradient (Fig. 2a) and a mean grain size of 6.5 mm (Fig. 2b). An edge crack of length $A_0$ ($A_0 \approx 0.7\ L$) was cut and tip-sharpened in each ice specimen. The response of the ice was monitored by using a number of surface-mounted linear variable
differential transducers (LVDTs). LVDTs were placed at five different locations along the crack to measure directly the crack opening displacements. Fig. 1 labels these positions as CMOD, COD, NCOD1, NCOD2, and NCOD3 for the crack mouth, intermediate crack, 10 cm behind the initially sharpened tip, 10 cm ahead of the tip, and 20 cm ahead of it, respectively. A hydraulically operated device was inserted in the mouth of the pre-crack to load the specimen, with a contact loading length of 150 mm, denoted by $D$ in Fig. 1. The tests were load controlled by a computer-operated closed-loop system that also recorded
the displacement measurements. The experiments were conducted with warm ice, -2 °C at the top surface as shown in Fig. 2a. The global behavior of the crack propagation was straight through the gauges. Detailed description of the experimental setup, ice growth, microstructure, and fractographic analysis is provided in (Gharamti et al., in-press).

### 2.2    Creep-recovery and monotonic loading profile

In two tests, ice specimens were subjected to creep-recovery loading followed by a monotonic fracture ramp. The creep-
recovery sequences consisted of four constant load applications, separated by zero load recovery periods. Each sequence was composed of alternating load/hold and release/recovery periods. Creep phases were applied at load levels of 0.4 kN, 0.8 kN,





1.2 kN, and 0.4 kN, as given by the loading signal in Fig. 3a. The loads were chosen low enough to avoid crack propagation and failure of the specimen. Each load-hold-unload was applied in the form of a trapezoidal wave function to avoid instantaneous load jump and drop; the load up was applied in approximately 10 seconds and released in approximately 10 seconds. The

slopes of the wave on load up and load release were 0.04kN/s, 0.08 kN/s, and 0.12 kN/s for the 0.4kN, 0.8 kN, and 1.2 kN load levels, respectively.

Once at the desired hold level, the load was kept constant for a predetermined time interval. The load intervals were multiple of the hold interval for the 0.4 kN load level, $\Delta t_1$ = 126 seconds. For the 0.8 kN and 1.2 kN load levels, the time interval was doubled and quadrupled: $2\Delta t_1$ = 252 seconds and $4\Delta t_1$ = 504 seconds, respectively. The four zero load recovery periods,

separating the creep load periods, were also function of $\Delta t_1$. Three recovery periods were held at zero load level for $5\Delta t_1$ = 630 seconds, but the last recovery period was maintained for a longer interval of $10\Delta t_1$ = 1260 seconds.

Immediately following the creep and recovery loading sequences, the specimen was loaded monotonically to failure on a load-controlled linear ramp. The ramp up to the peak load and unloading were each applied over an interval of $\Delta t_1$.

### 2.3    Cyclic-recovery and monotonic loading profile

In three tests, ice specimens were loaded with cyclic-recovery sequences followed by a fracture ramp, as shown in Fig. 3b. The cyclic-recovery loading consisted of 3 sequences, each being composed of four fluctuating loads, at the levels of 0.4 kN, 0.8kN, and 1.2 kN. Each cyclic sequence continued for a constant time interval $\Delta t_2$ = 480 seconds. The slopes of the wave on the load up and load release were 1/150 kN/s, 1/75 kN/s, and 1/50 kN/s for the 0.4kN, 0.8kN, and 1.2 kN load levels, respectively. The 0.4kN, 0.8kN, and 1.2 kN cyclic load periods were followed by zero load recovery periods of $1.25\Delta t_2$ = 600 seconds, $1.25\Delta t_2$

= 600 seconds, and $2.5\Delta t_2$ = 1200 seconds, respectively.

At the completion of the cyclic-recovery loading sequences, the specimen was loaded to failure by a monotonic linear ramp. The ramp up to the peak load and unloading were each applied over an interval of $0.25\Delta t_2$ = 120 seconds.

### 3    Nonlinear time-dependent modeling of S2 columnar freshwater ice

The model applied in this section to characterize the nonlinear viscoelastic/viscoplastic response of S2 columnar freshwater

ice was presented by Schapery; it was used to model the time-dependent mechanical response of polymers in the nonlinear range under uniaxial stress-strain histories (Schapery, 1969). Schapery's stress-strain constitutive equations are derived from nonlinear thermodynamic principles, and are very similar to the Boltzmann superposition integral form of linear theory (Flügge, 1975). Schapery's model represents the material as a system of an arbitrarily large number of nonlinear springs and dashpots.

The equations in this section are presented in terms of load and displacement instead of the original stress-strain relations.

The notations of the original equations in (Schapery, 1969) are modified to bring out similarity between all the equations in the paper.

When the applied loads are low enough, the material response is linear. For an arbitrary load input, $P = P(t)$ applied at $t = 0$, Boltzmann's law approximates the load by a sum of a series of constant load inputs and describes the linear viscoelastic





displacement response of the material using the hereditary integral in a single integral constitutive equation. The Boltzmann

superposition principle states that the sum of the displacement outputs resulting from each load step is the same as the displacement output resulting from the whole load input. If the number of steps tends to infinity, the total displacement is given as:

$$\delta(t) = C_0 P + \int_0^t \Delta C(t - \tau) \frac{dP}{d\tau} d\tau, \tag{1}$$

where $C_0$ is the initial, time-independent compliance component and $\Delta C(t)$ is the transient, time-dependent component of

compliance.

Turning now to nonlinear viscoelastic response, Schapery developed a simple single-integral constitutive equation from nonlinear thermodynamic theory, with either stresses or strains entering as independent variables (Schapery, 1969). Using load as the independent variable, the displacement response under isothermal and uniaxial loading takes the following form:

$$\delta(t) = g_0 C_0 P + g_1 \int_0^t \Delta C(\psi - \psi') \frac{d(g_2 P)}{d\tau} d\tau, \tag{2}$$

where $C_0$ and $\Delta C$ are the previously defined components of Boltzmann principle, $\psi$ and $\psi'$ are the so-called reduced times defined by:

$$\psi = \int_0^t \frac{dt'}{a_P} \quad \text{and} \quad \psi' = \psi(\tau) = \int_0^\tau \frac{dt'}{a_P} \tag{3}$$

and $g_0, g_1, g_2,$ and $a_P$ are nonlinear functions of the load. Each of these functions represents a different nonlinear influence on the compliance: $g_0$ models the elastic response, $g_1$ the transient response. $g_2$ the loading rate, and $a_P$ is a time scale shift factor.

These load-dependent properties have a thermodynamic origin. Changes in $g_0, g_1,$ and, $g_2$ reflect third and higher order stress-dependence of the Gibbs free energy, and changes in $a_P$ are due to similar dependence of both entropy production and the free energy. These functions can also be interpreted as modulus and viscosity factors in a mechanical model representation. In the linear viscoelastic case, $g_0 = g_1 = g_2 = a_P = 1$, and Schapery's constitutive equation (2) reduces to Boltzmann's equation (1).

Equation (2) contains one time-dependent compliance property, from linear viscoelasticity theory, $\Delta C$ and four nonlinear

load-dependent functions $g_0, g_1, g_2,$ and $a_P$, which reflect the deviation from the linear viscoelastic response, that need to be evaluated. Schapery's model uses experimental data to evaluate the material property functions in (2). Lou and Schapery outlined a combined graphical and numerical procedure to evaluate these functions (Lou and Schapery, 1971). In their work, a data-reduction method was applied to evaluate the properties from the creep and recovery data. Papanicolaou et al proposed a method capable of analytically evaluating the material functions using only limiting values of the creep-recovery test (Papan-

icolaou et al., 1999). Numerical methods are also employed and are the most commonly used techniques; they are based on fitting the experimental data to the constitutive equation (LeClair et al., 1999). In the current study, a numerical-experimental





procedure is adopted. An optimization procedure is applied using the Nelder-Mead (N-M) method (Nelder and Mead, 1965) to back-calculate the values that achieve the best fit between the model and the experimental data. To avoid multiple fitting treatments of data and account for the mutual dependence of the functions, the properties were determined from the full data. This

avoided errors that may result from separating the data into parts and estimating the functions independently from different parts.

Schapery later updated his formulation (Schapery, 1997). He added a viscoplastic term to account for the viscoplastic response of the material and stated that the total compliance can be represented as the summation of elastic, viscoelastic, and viscoplastic components. Adamson and Dempsey applied Schapery's updated constitutive equation to model the crack mouth

opening displacement of saline ice in an experimental setup similar to the current study (Adamson and Dempsey, 1998). The theory represents the displacement at the crack mouth ($\delta_{\text{CMOD}}$) as the sum of elastic, viscoelastic, and viscoplastic components:

$$\delta_{\text{CMOD}} = \delta^e_{\text{CMOD}} + \delta^{ve}_{\text{CMOD}} + \delta^{vp}_{\text{CMOD}} \tag{4}$$

where

$$\delta^e_{\text{CMOD}} = g_0 C_e P \tag{5}$$

$$\delta^{ve}_{\text{CMOD}} = g_1 \int_0^t C_{ve}(\psi - \psi') \frac{d(g_2 P)}{d\tau} d\tau \tag{6}$$

$$\delta^{vp}_{\text{CMOD}} = C_{vp} \int_0^t g_3 P d\tau \tag{7}$$

In the above equations, $\psi$ and $\psi'$ are defined in (3). $g_0, g_1, g_2, g_3,$ and $a_P$ are nonlinear load functions to be determined. The coefficients $C_e, C_{ve},$ and $C_{vp}$ are the elastic, viscoelastic, and viscoplastic compliances, respectively. Schapery's equation has been developed for uniaxial loading. The experimental problem at hand is not precisely uniaxial, but it is approximated as so, and Schapery's equations are used to analyze the experimental data. Few assumptions are applied at this point and are based on the choices made in (Adamson and Dempsey, 1998). For ice, the elastic displacement is linear with load; this immediately leads

to $g_0 = 1$. Schapery stated that $g_1 = a_P = 1$ if the instantaneous jump and drop in the displacement are equal (Schapery, 1969). Examination of the current data shows that this condition is not valid, and the functions need to be evaluated. Accordingly, the following approximations are employed:

$$g_1 \propto P^a; \quad g_2 \propto P^{b-1}; \quad g_3 \propto P^{c-1}; \quad a_P \propto P^d \tag{8}$$

The viscoelastic compliance is assumed to follow a power law in time with a fractional exponent $n$. This gives:

$$C_{ve}(t) \approx \kappa t^n \tag{9}$$





Incorporating each of these conditions, the total displacement is expressed as

$$\delta_{\text{CMOD}} = C_e P + \kappa P^a \int_0^t t^n \frac{(t-\tau)}{P^d} \frac{dP^b}{d\tau} d\tau + C_{vp} \int_0^t P^c d\tau \tag{10}$$

where $\delta_{\text{CMOD}}$, $P$ and $t$ are in m, N, and seconds, respectively. It follows from (10) that two unknown parameters ($C_e$, and $C_{vp}$), one unknown constant ($\kappa$), and five unknown exponents ($a, b, c, d,$ and $n$) need to be determined. As previously mentioned, the problem is formulated as a least-squares problem and optimized through the N-M technique, by minimizing the objective function $\mathcal{F}$ given by the difference between the model and data, as shown in (11). The components of the total displacement were computed and optimized using MATLAB. Initial guesses of the exponents on the load and time functions were assumed based on previous work on saline ice. The optimized values were then obtained by comparing the model response and the experimentally measured response over the full length of the test up to crack growth initiation.

$$\mathcal{F} = \underset{C_e, C_{vp}, a, b, \ldots}{\arg\min} \sum_{i=1}^N \left\| M_i(C_e, C_{vp}, \kappa, a, b, c, d, n) - D_i \right\|_2 \tag{11}$$

where $M_i$ and $D_i$ refer to the CMOD values given by the model (10) and the experimental data, respectively. $\|.\|_2$ is the Euclidean norm of a vector. $N$ is the number of data points ($\approx$ 2e6 points).

As mentioned earlier, Schapery's model originated from the thermodynamic theory. The model is not physically-based, and its parameters are not linked to the microstructural properties of the ice (dislocation density, grain size, ...). In addition, the analysis does not account for the formation of fracture process zone in the vicinity of the crack tip. Schapery's formulation models the experimental response until crack growth initiation and does not account for crack propagation.

## 4 Results and discussions

### 4.1 Experimental results

This section presents the results measured and computed for the LC tests. The current results are compared with the fracture results of monotonically loaded DC tests of the same ice and same specimen size (3m x 6m) (Gharamti et al., in-press). The main aim is to elucidate the effect of creep and cyclic sequences on the fracture properties.

#### 4.1.1 Effect of the creep and cyclic sequences on the fracture properties

Table 1 shows the measured and computed parameters for the LC experiments. $t_f$ represents the time to failure, computed from the fracture ramp. From the failure load and dimensions, an apparent fracture toughness ($K_Q$) is computed using the weight function procedure outlined in Section 4 of (Gharamti et al., in-press). CMOD is computed at crack growth initiation. $\dot{\text{CMOD}}$ indicates the displacement rate at the crack mouth and is obtained by dividing CMOD by the failure time. Similarly, NCOD1 (see Fig. 1) represents the displacement at crack growth initiation near the initially sharpened crack tip. $\dot{\text{NCOD1}}$ indicates the displacement rate in the vicinity of the tip and is obtained by dividing NCOD1 by the failure time.



Fig. 4 gives the results of the apparent fracture toughness $K_Q$, crack mouth opening displacement CMOD, and near crack-tip opening displacement NCOD1 as a function of the loading rate for the DC tests (Gharamti et al., in-press) and the current LC tests. In these subplots, first-order power-law fits were applied to the data of the DC tests. The LC values lie above, below, and along the DC fit. No clear effect of creep and cyclic loading on the fracture properties was detected.

Figs. 5a and 5b show the experimental load versus the crack opening displacement at the crack mouth for the DC and the LC tests, respectively. Fig. 5c displays a zoomed view of the fracture ramp of the LC tests. Comparing the failure loads of the DC and LC tests indicates that the failure loads, of tests with comparable loading rates, were similar. Therefore, in these experiments, the creep and cyclic sequences had no influence on the failure load.

Table 1 presents several elastic moduli for each test. The elastic moduli were calculated from the load-CMOD record following Section 4 of (Gharamti et al., in-press). For the creep tests (RP15 and RP16), this procedure is repeated for the four creep cycles, resulting in $E_1$, $E_2$, $E_3$, $E_4$, and for the fracture ramp, resulting in $E_f$. Similarly for the cyclic tests (RP17, RP18, and RP19), the moduli calculation was done for the last cycle of each cyclic sequence, giving steady state moduli $E_1$, $E_2$, $E_3$, and for the fracture ramp, resulting in $E_f$. Some of the values are missing, caused by the fact that the initial portion of the associated load-CMOD curve was very noisy. The values of the elastic moduli calculation for the the creep/cyclic sequences and fracture ramps were similarly linear upon load application, as shown by the loading slope in Figs. 5c, 6a, and 6b. This linearity justifies the choice of $g_0 = 1$ in the elastic CMOD component in Eq. (5).

Table 1 in (Gharamti et al., in-press) presents the elastic modulus ($E_{\mathrm{CMOD}}$) calculated at the crack mouth for the DC tests; $E_{\mathrm{CMOD}}$ is similar to $E_f$ in Table 1 here; both values lie within the same range. Therefore, the creep and cyclic sequences preceding the fracture ramp did not affect the load-CMOD prepeak behavior. However, the sequences affected the post-peak response as can be distinguished from Fig. 5b which displays more decay behavior than Fig. 5a. The gradual decay of the load portrays the time dependency in the behavior of freshwater ice.

### 4.1.2 Ice response under the testing conditions

Fig. 7 shows the experimental results for RP16: the applied load and the crack opening displacements at the crack mouth (CMOD), halfway of the crack (COD), and 10 cm behind the tip (NCOD1) (see Fig. 1). Similarly, Fig. 8 shows the experimental response for RP17. The time-dependent nature of of the ice response is evident. A complete load-CMOD curve was obtained during loading and unloading for each test of Table 1, indicating stable crack growth.

It is clear from Figs. 7b and 8b that the CMOD, COD, and NCOD1 displacements were composed mainly of elastic and viscoplastic components. No viscoelasticity was detected in the displacement-time records for all the tests. The primary (transient) creep stage was absent. The load sequences were characterized by a non-decreasing displacement rate at all levels. The displacement-time slope was linear and constant, indicating that the secondary/steady-state creep regime dominated during each load application. Although the recovery time was longer than the loading time, $\geq 1.25\Delta t_1$ (Creep test, Fig. 3a and Section 2.2) and $\geq 1.25\Delta t_2$ (Cyclic test, Fig. 3b and Section 2.3), the recovery (unload) phases consisted of a small elastic recovery (instantaneous drop) and unrecovered viscoplastic displacement. The behavior as observed resembles the response of





a simple Maxwell model composed of a series combination of a nonlinear spring and nonlinear dashpot. There is no delayed elastic recovery, but there is the elastic response and a permanent deformation.

Figs. 6a and 6b support the same analysis. Unlike the viscoelastic response (Fig. 6c) which displays no residual displacement
in the loading and unloading hysteresis diagram, the current load-CMOD plots showed large permanent displacement after each loading cycle.

Interestingly, the ice behavior in the current study differs from previous experimental creep and cyclic work on freshwater ice. Large delayed elastic or recoverable component has been previously observed. Several researchers performed creep experiments at lower temperatures (Mellor and Cole, 1981, 1982, 1983; Cole, 1990; Duval et al., 1991) and reported consid-
erable recovery. Duval conducted torsion creep tests on glacier ice at a similar testing temperature of -1.5 °C (Duval, 1978). When unloaded, the ice exhibited creep recovery. According to his analysis: during loading, the internal stresses opposing the dislocation motion increases; upon unloading, the movement of dislocations produced the reversible deformation and is caused by the relaxation of internal stresses. Sinha (Sinha, 1978; Sinha et al., 1979) concluded that high-temperature creep of polycrystalline ice is associated with grain boundary sliding. Cole developed a physically-based constitutive model in terms of
dislocation mechanics (Cole, 1995) and quantified two mechanisms of anelasticity: dislocation and grain boundary relaxations. He demonstrated that the increased temperature sensitivity of the creep properties of ice within a few degrees of the melting point is due to a thermally induced increase in the dislocation density (Cole, 2020). The question then arises as to why columnar freshwater ice tested at -2 °C did not show a delayed elastic effect, and the microstructural changes were mainly irreversible upon unloading?

An influencing factor is the mechanisms taking place in the process zone. There is possibility that the loading conditions produced dislocations which would ordinarily generate some viscoelastic deformation upon unloading. However, local damage in the process zone relieved the internal stresses that are needed to drive the dislocation recovery. Thus, any microstructural damage that occurred during loading manifested as permanent deformation at the end of the test. It is noteworthy that the earlier studies used test sizes which are smaller than the plate size used here. It was shown in the DC fracture tests (Gharamti et al.,
in-press) that scale had an effect at the tested loading rates. It is probable that the specimen size influenced the time-dependent deformation of freshwater ice. When the specimen dimensions are several meters, apparently viscoelasticity is not an important deformation component.

The ice response indicates that the combined effects of the geometry, the applied loading profiles, the warm temperature (-2 °C), and the testing conditions triggered an instantaneous transformation from the primary stage to the steady-state stage,
resulting in permanent irreversible deformation that accumulated after each creep/cyclic-recovery sequence. This concludes that the response of columnar freshwater S2 ice in these tests is elastic-viscoplastic.

## 4.2 Schapery-optimization modelling analysis

The nonlinear theory, outlined in Section 3, was used to analyze the experiments. The results of the initial optimization trials confirmed the previous analysis; the viscoelastic component $\delta_{\text{CMOD}}^{ve}$ had no effect on the final fit between the data and the





model. The variation of the constants corresponding to $\delta_{\text{CMOD}}^{ve}$, $\kappa, a, b, d,$ and, $n$ didn't affect the final converged values of $C_e, C_{vp},$ and $c$.

The final optimization runs were carried out by considering the elastic and viscoplastic components (first and last terms of Eq. (10)) only. This resulted in 2 parameters, $C_e$ and $C_{vp}$, and one exponent $c$, that need to be optimized. The optimization converged results are given in Table 2: $C_e, C_{vp}$ and $c$. For all the tests, the %reduction of the objective function exceeded

95% and about 110 iterations were needed to reach convergence. A value of $c = 1$ for the viscoplastic load function provided the best fit between the model and the experiment at all load levels over the total experimental time up to the peak load. The final compliance values of the elastic and viscoplastic components were in the ranges 1.8-3.8 $\times 10^{-8}$mN$^{-1}$ and 0.2-1 $\times 10^{-10}$mN$^{-1}$s$^{-1}$. respectively.

Figs. 9 and 10 give the model results, obtained using Eqs. (4-10), and the experimental results for experiments RP16 and

RP17, respectively. Figs. 9a and 9b show the measured load and the load applied to the model and the measured CMOD-time record compared to the response of the model, respectively for RP16. Figure 10 shows similar plots for experiment RP17. Test RP17 showed an excellent model-experiment fit for the three cyclic-recovery sequences over the load and unload periods. The recovery displacements were fitted accurately by the model in the three recovery periods. The experimental response for the creep-recovery test RP16 appeared to conform to the model results, but the model under estimated the recovery displacement

in the first two cycles with a maximum misfit of $\approx 2e^{-5}$ $\mu$m. Schapery's model has been tested for creep-recovery sequences of saline ice with an increasing load profile (Schapery, 1997; Adamson and Dempsey, 1998; LeClair et al., 1999, 1996). This is the first application of the model with a load profile of increasing and decreasing load levels (Fig. 3a). The model succeeded to follow the increasing and decreasing load levels and the corresponding recovery phases. The model generated the peak displacement values for all load levels with a misfit of $1e^{-5}$ $\mu$m for the last two creep cycles. The observed misfit (1-2 $e^{-5}$

$\mu$m) is small and should be ignored. It is related to the accuracy of the measurement line (LVDT + amplifier + data processing unit) which is affected by many environmental and technical factors. Thus, the implemented model provided a good fit with the data over the creep-recovery and cyclic-recovery sequences.

Considering the fracture ramp, Schapery's nonlinear equation succeeded to model the monotonic displacement response up to crack growth initiation perfectly well for all the tests. As previously mentioned, the model does not account for crack

propagation, so modeling was applied until the peak load. The model was also successful in predicting the critical crack opening displacement values at the failure load. Thus, the model gives a very close prediction of the experimental data over the whole loading profile up to the failure load. The other tests displayed the same experiment-model agreement.

In this study, Schapery's constitutive model is tested for the first time for freshwater ice. The match between the model and the measured data, over the creep/cyclic-recovery sequences and fracture monotonic ramp, provides a firm support of the

ability of Schapery's constitutive model to describe the time-dependent response of columnar freshwater S2 ice up to crack growth initiation. Figs. 11a and 11b show the contribution of each individual model components, elastic and viscoplastic, to the total CMOD displacement, for RP16 and RP17, respectively. As mentioned earlier, the elastic and viscoplastic components account for the total deformation. For RP16, the viscoplastic component dominated over the elastic component. For RP17, the elastic and viscoplastic components contributed equally to the total displacement.




The applicability of the proposed model and the fitted parameters are limited to the studied ice type, geometry, specimen size, ice temperature, and the current testing conditions. Variation in the operating conditions will change the dominant deformation mechanisms and the ice behavior; and accordingly, new model parameters are needed to adapt to the new response.

## 5   Conclusions

In the present work, five 3m x 6m freshwater S2 ice specimens were tested under creep/cyclic-recovery sequences followed by a monotonic ramp at -2°C. The tests were load controlled and led to complete fracture of the specimen. The purpose of this study was to examine the time-dependent behavior of freshwater ice using a joint experimental-modeling approach.

In the experimental part, the tests aimed to (1) measure and examine the time-dependent response of columnar freshwater S2 ice through the applied creep/cyclic-recovery sequences and (2) investigate the effect of creep and cyclic sequences on the fracture parameters/behavior through the fracture monotonic ramp. The current tests were compared with other monotonically loaded tests, and fracture parameters were computed and analyzed. The results showed that the creep and cyclic sequences had no clear effect on the apparent fracture toughness, the failure load, and the crack opening displacements. The ice response at the testing conditions was elastic-viscoplastic, and the steady-state regime dominated the loading phases. The conducted experiments provided a novel observation for the time-dependent behavior of freshwater ice. Though the delayed elastic component has been reported as a major creep component (Duval, 1978), no viscoelasticity was detected in this study. The primary (transient) creep stage was absent. This indicates that the collective effects of the testing conditions and plate configuration triggered an instantaneous transformation to the steady-state regime resulting in permanent (unrecoverable) displacement.

In the modeling part, Schapery's nonlinear constitutive model was applied for the displacement response at the crack mouth. The elastic-viscoplastic formulation succeeded to predict the experimental response of columnar freshwater S2 ice over the applied loading profile up to crack growth initiation. The model parameters were obtained via an optimization procedure using the N-M method by comparing the model and experimental CMOD values.

The proposed model parameters are valid only for the studied ice type, geometry, specimen size, ice temperature, and the range of applied load experienced in the experiments. Schapery's model was selected in this study, as it is able to capture the sort of time dependent behavior known to occur in ice and produces a simple and expedient way to help understand the observed behavior. More thorough analysis with a physically based approach is left to the future.

*Code and data availability.* The code used for material modeling is written in Matlab. Scripts used for analysis and more detailed information of the experimental results are available from the authors upon request.

*Author contributions.* All authors designed the study and performed the experiments. I.E.G. generated the results and drafted the paper. All authors commented on the text.



*Competing interests.* The authors declare that they have no conflict of interest.

*Acknowledgements.* This work was funded though the Finland Distinguished Professor programme "Scaling of Ice Strength: Measurements and Modeling", and through the ARAJÄÄ research project, both funded by Business Finland and the industrial partners Aker Arctic Technology, Arctech Helsinki Shipyard, Arctia Shipping, ABB Marine, Finnish Transport Agency, Suomen Hyötytuuli Oy, and Ponvia Oy. This financial support is gratefully acknowledged. The authors would like to thank Dr. David Cole for taking the time and effort to review the manuscript. The first author (I.E.G.) is thankful to Dr. Kari Santaoja for useful and enlightening discussions. The second author (J.P.D.)
thanks Business Finland for support by the FiDiPro Professorship from Aalto University, and the sabbatical support from Aalto University, which collectively supported an annual visit 2015-2016, and summer visits 2017-2019.



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





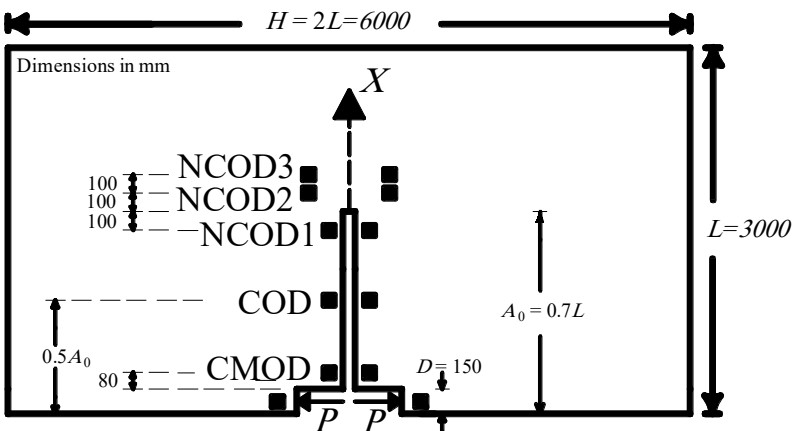

**Figure 1.** (a) Specimen geometry, edge cracked rectangular plate of length $L$, width $H$, and crack length $A_0$.





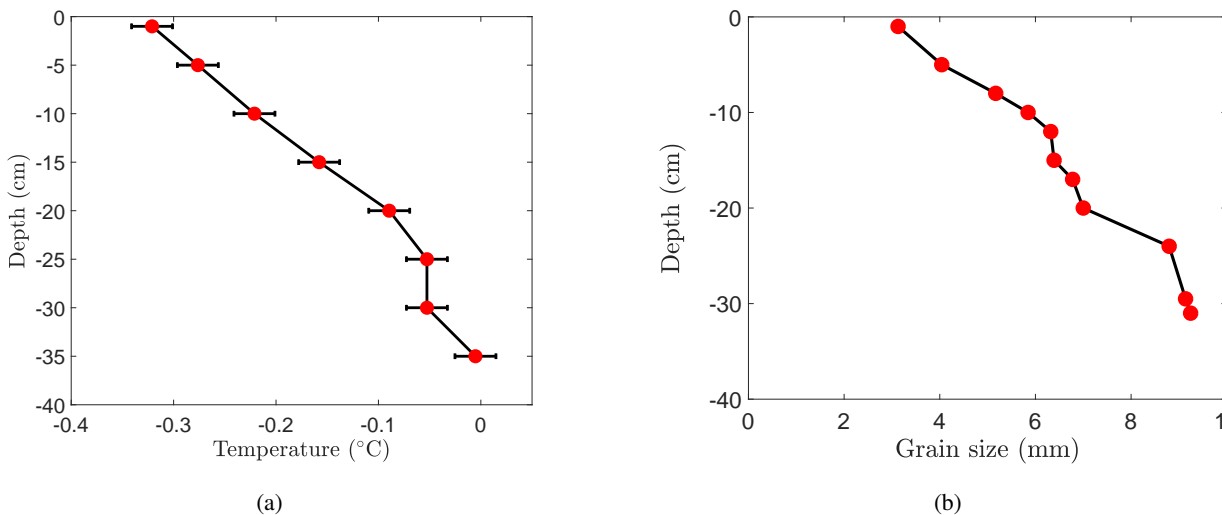

(a)                                                (b)

**Figure 2.** (a) Temperature profile. Each data point represents the average of measurements taken at the same depth of different ice cores throughout the one month duration of the test program. (b) Grain size distribution. Each data point is measured from one thin section.





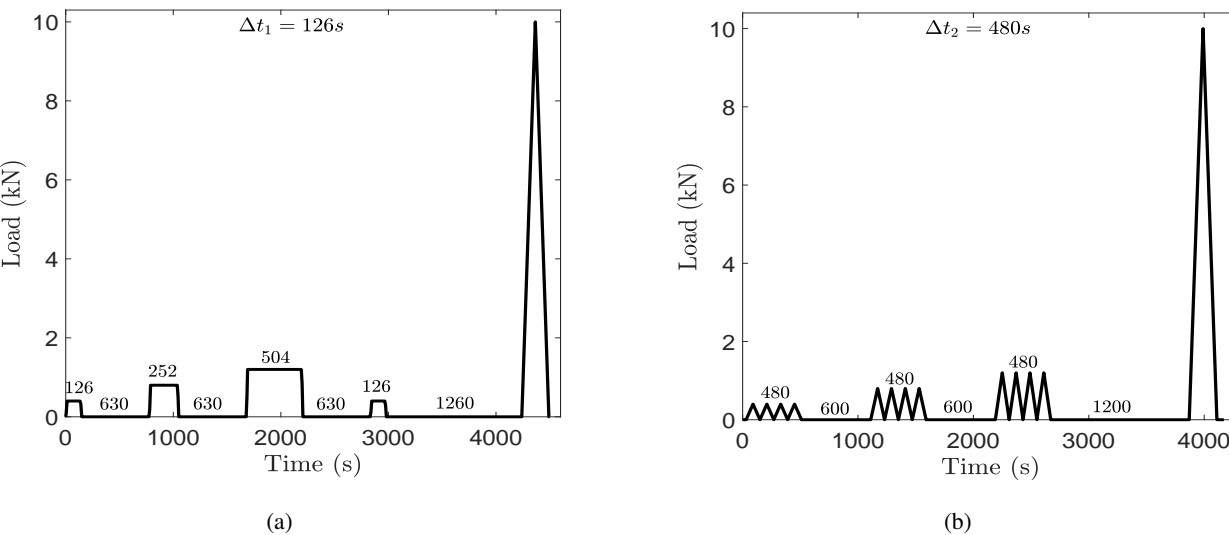

(a)                                             (b)

**Figure 3.** Loading consisting of (a) creep-recovery and (b) cyclic sequences followed by a monotonic fracture ramp. The number above each segment indicates the duration in s.







**Figure 4.** Experimental results for the (a) apparent fracture toughness $K_Q$, (b) crack mouth opening displacement CMOD and (c) near crack tip opening displacement NCOD1 at crack growth initiation, as a function of loading rate for the monotonically-loaded DC tests Gharamti et al. (in-press) and the creep/cyclic and monotonically-loaded LC tests.





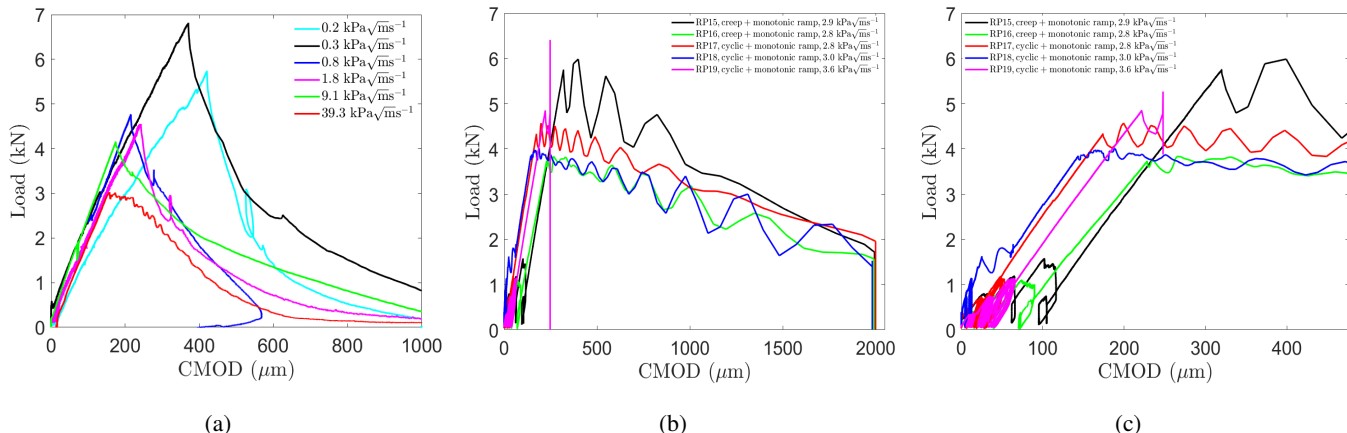

**Figure 5.** Measured load versus CMOD for the (a) DC tests Gharamti et al. (in-press), (b) LC tests, and (c) LC tests up to the peak load.



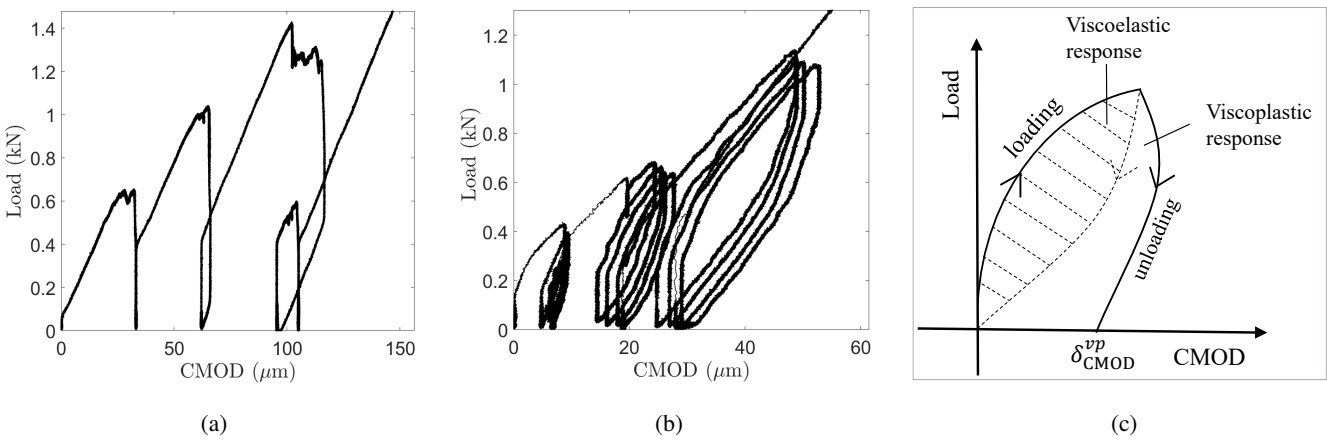

|     |     |     |
| --- | --- | --- |
| (a) | (b) | (c) |

**Figure 6.** Load versus CMOD over the (a) creep-recovery cycles for RP15 and the (b) cyclic-recovery sequences for RP17. (c) Schematic illustration of the hysteresis load-displacement diagram. The whole of the hysteresis loop area is the energy loss per cycle. The dashed area is the part of that total that is due to the viscoelastic mechanism and the rest is due to viscous processes.





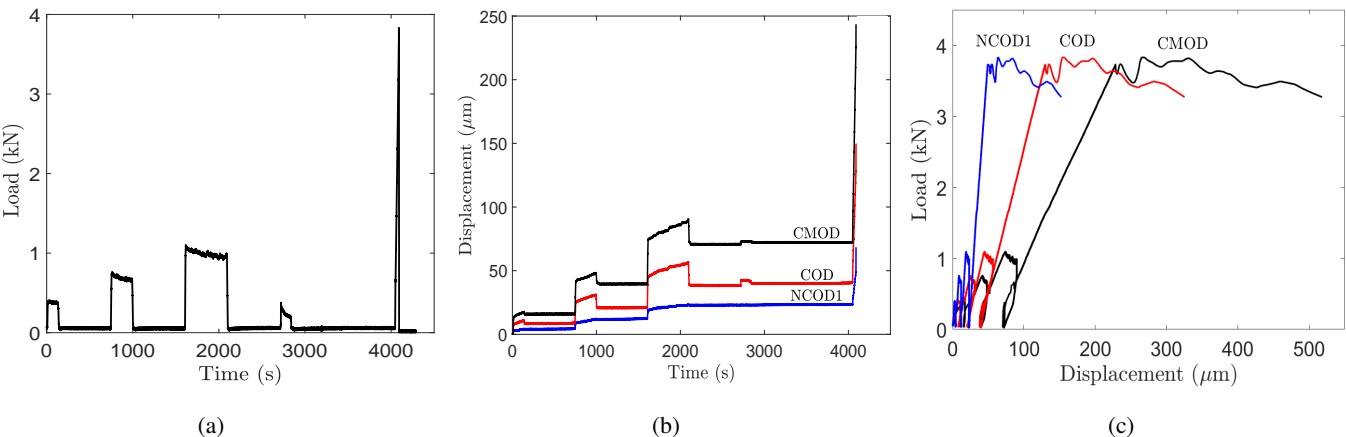

(a)             (b)             (c)

**Figure 7.** Experimental results for RP16. (a) Load at the crack mouth, see Fig. 1. (b) Displacement - time records. (c) Load - displacement record.





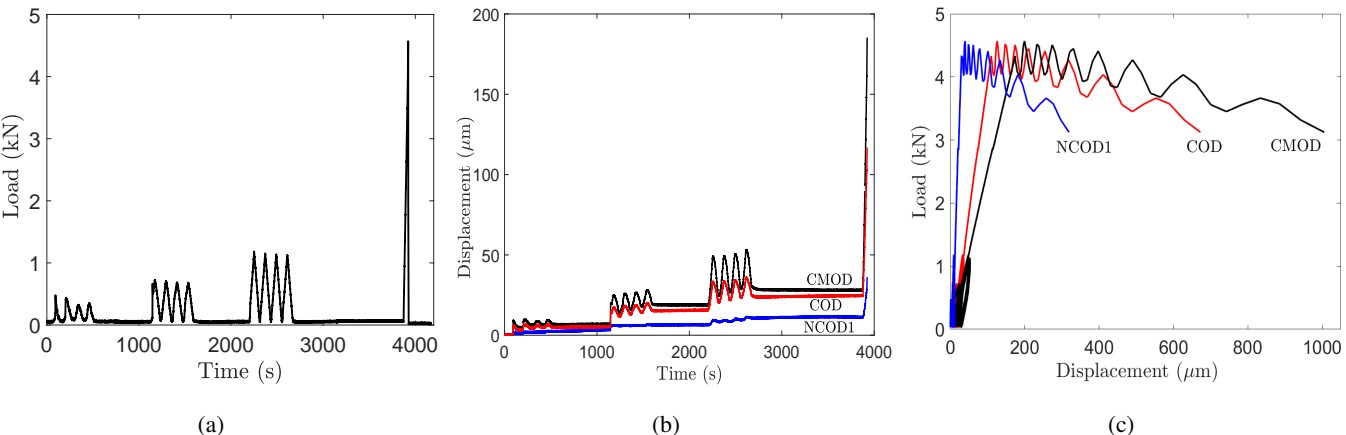

|     |     |     |
| --- | --- | --- |
| (a) | (b) | (c) |

**Figure 8.** Experimental results for RP17. (a) Load at the crack mouth, see Fig. 1. (b) Displacement - time records. (c) Load - displacement record.





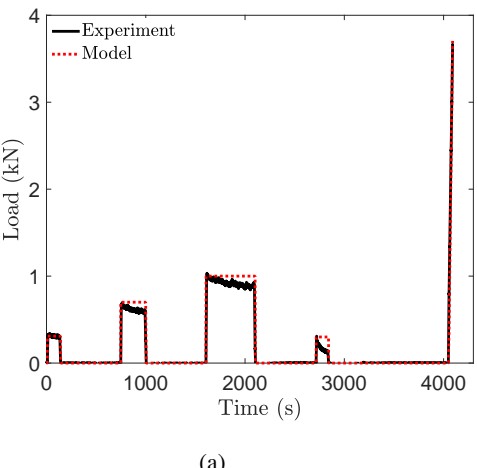

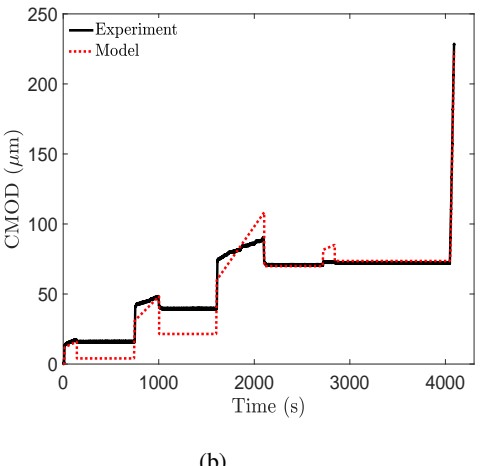

(a)              (b)

**Figure 9.** Experimental and model results for RP16. (a) Load at the crack mouth, see Fig. 1. (b) CMOD - time records.





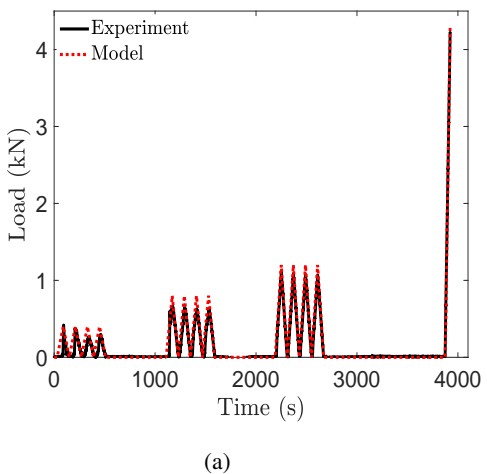
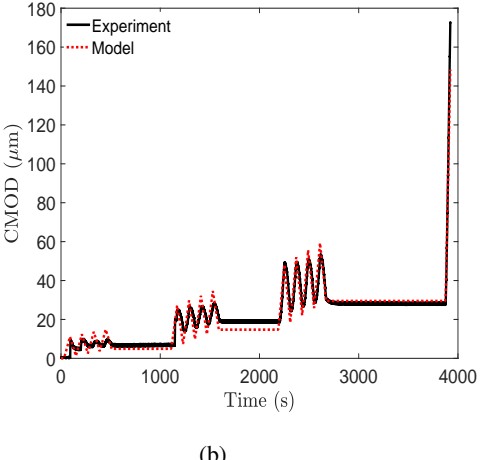

(a)                                          (b)

**Figure 10.** Experimental and model results for RP17. (a) Load at the crack mouth, see Fig. 1. (b) CMOD - time records.





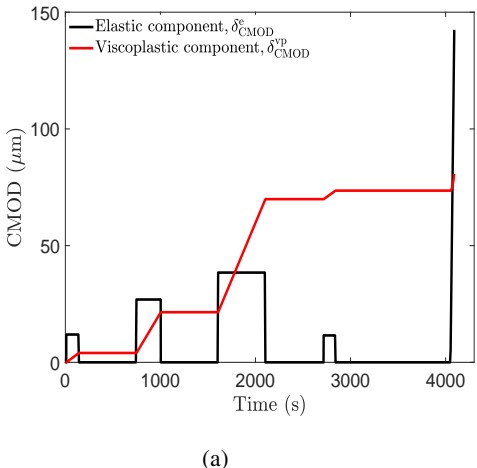

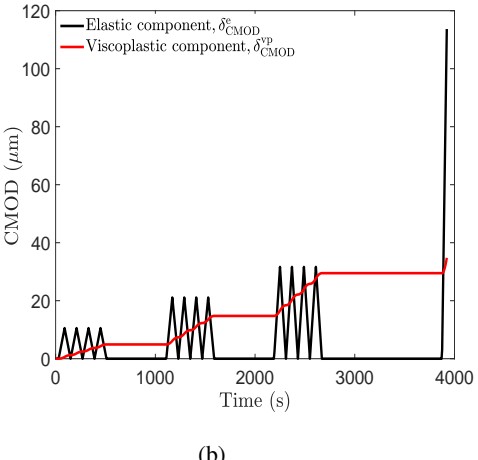

(a)            (b)

**Figure 11.** Contribution of each individual model component to the total CMOD displacement for (a) RP16 and (b) RP17.

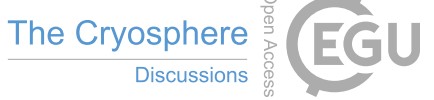



**Table 1.** Measured experimental data and computed results for the LC tests.

| Test | Type | $L$ | $H$ | $A_0$ | $h$ | $E_1$ | $E_2$ | $E_3$ | $E_4$ | $E_f$ | $P_{max}$ | $t_f$ | $K_Q$ | $\dot{K}$ | CMOD | $\dot{\mathrm{CMOD}}$ | NCOD1 | $\dot{\mathrm{NCOD1}}$ |
|------|------|-----|-----|-------|-----|-------|-------|-------|-------|-------|-----------|-------|-------|-----------|------|------|-------|-------|
| | | (m) | (m) | (m) | (mm) | (GPa) | (GPa) | (GPa) | (GPa) | (GPa) | (kN) | (s) | (kPa$\sqrt{\mathrm{m}}$) | (kPa$\sqrt{\mathrm{m}}\mathrm{s}^{-1}$) | ($\mu$m) | ($\mu$m s$^{-1}$) | ($\mu$m) | ($\mu$m s$^{-1}$) |
| RP15 | creep | 3 | 6 | 2.1 | 364 | 6.6 | 6.7 | 7.3 | 7.4 | 6.9 | 6.0 | 68.2 | 198.6 | 2.9 | 320.1 | 4.7 | 53.6 | 0.8 |
| RP16 | creep | 3 | 6 | 2.1 | 385 | 5.6 | 5.8 | 7.6 | - | 6.0 | 3.8 | 42.8 | 120.2 | 2.8 | 228.2 | 5.3 | 49.1 | 1.1 |
| RP17 | cyclic | 3 | 6 | 2.1 | 407 | 6.5 | - | 7.6 | - | 6.6 | 4.6 | 49.3 | 135.6 | 2.8 | 173.7 | 3.5 | 30.0 | 0.6 |
| RP18 | cyclic | 3 | 6 | 2.1 | 408 | - | - | - | - | 5.3 | 4.0 | 40.1 | 118.7 | 3.0 | 143.7 | 3.6 | 28.5 | 0.7 |
| RP19 | cyclic | 3 | 6 | 2.1 | 412 | - | 7.0 | 6.6 | - | 6.3 | 6.4 | 52.5 | 187.9 | 3.6 | 221.4 | 4.2 | 44.0 | 0.8 |





**Table 2.** Optimization results computed using Schapery's model.

| Test | $\dot{K}$ | $C_e$ x $10^8$ | $C_{vp}$ x $10^{10}$ | $c$ |
|------|-----------|----------------|----------------------|-----|
|      | (kPa$\sqrt{\text{m}}$s$^{-1}$) | (mN$^{-1}$) | (mN$^{-1}$s$^{-1}$) | |
| RP15 | 2.912 | 3.330 | 1.061 | 1 |
| RP16 | 2.808 | 3.845 | 0.974 | 1 |
| RP17 | 2.750 | 2.637 | 0.512 | 1 |
| RP18 | 2.960 | 1.861 | 0.209 | 1 |
| RP19 | 3.579 | 2.775 | 0.938 | 1 |