# Peer review of "Creep and fracture of warm columnar freshwater ice"

_The Cryosphere, 2020_

## Referee Comment (RC1) · Anonymous Referee #1 · 6 Oct 2020

This manuscript describes the results of experiments on the deformation of freshwater ice near its melting point. Specifically, it describes the rate of opening of notches that were cut into large blocks ( $\sim$0.4 m x 3 m x 6 m) of columnar-grained ice that possessed the S2 growth texture. The blocks were floating in an ice tank and the sides of the notch were pried apart at different rates of loading until the ice split. Prior to fracture, also described, load was applied either cyclically or constantly for a period of time. The principal finding and conclusion is that under the conditions of the experiment, "creep and cyclic sequences had no clear effect on the apparent fracture toughness, the failure load, and the crack opening displacements" (line 320-321). Although a negative conclusion, that result is worth publication in The Cryosphere.

The authors go further, however, weakening/endangering the manuscript. They ana-

lyze their data in terms of a constitutive model that was developed by Schapery (1969, 1997) for uniaxial loading (noted in lines 171-173). But in the experiments at hand, deformation occurred under a multiaxial stress state. Given that the model and the data relate to different states of stress and given that values of the many (eight) unknown parameters in the model were derived by fitting the data and not from independent measurements, it is difficult to accept the statement (lines 303-305) that the analysis " provides a firm support of the ability of Schapery's constitutive model to describe the time-dependent response of columnar freshwater S2 ice up to crack growth initiation." It is even more difficult to accept the claim (lines 242-243) that under the conditions of these experiments "there is no delayed elastic recovery".

The title presents a problem: it is misleading. This is not the kind of experiment that allows a characterization of elastic-viscoelastic deformation of ice. Rather, as already noted, it allows a conclusion to be made on fracture toughness and its insensitivity to pre-strain. The title needs to be changed to reflect that finding.

The other problem is that the manuscript contains a contradiction. The claim that the experiments were performed on ice at -2 o C contradicts the temperature profile shown in Fig. 3a. There, where temperature is plotted versus depth (from $\sim$0 to 35 cm) in the ice, temperature ranges from -0.3 o C near the top to $\sim$ 0 o C near the bottom.

Finally, it would be helpful to know in which journal the repeated reference to Gharamti et al. is "in press".
* * *

---

## Referee Comment (RC2) · Anonymous Referee #2 · 18 Oct 2020

This article reports the experimental results from load controlled mode I fracture tests on columnar freshwater S2 ice at -2 C. To characterize the load-displacement behavior observed in the experiments, the authors use Schapery's constitutive model and establish its ability to describe the crack mouth opening displacements under creep and cyclic-recovery load sequences. The novelty/merit of this work relies on the uniqueness of the experiments, which are conducted on floating 3 m x 6 m rectangular plates in the Ice Tank at Aalto University. To my knowledge, very few laboratories in the world have the capability to conduct such experiments and so this type of experimental data is hardly available in the literature. From this perspective, I think the article is worthy of publication, however, I think the paper needs revision before it can be published. Below I provide my detailed comments on the manuscript.

[Figure]

Introduction

Page 1, line 14 – The eventual goal of these experiments seems to be to better describe or model ice loads during sea ice floe interactions with structures in the Arctic. As the authors mention the deformation modes are quite complex during ice-structure interaction, but there is no literature cited on this work. For example, Claude Daley at MUN has done experimental and modeling work on this using ice indentation experiments on structures, which seems to be the most relevant mechanism for transfer of ice loads onto structures. One would rarely imagine a floating ice flow to be subjected to mode I tension. Please add more description to the introduction about what experimental data is available, what motivated the current experimental study and why mode I fracture experiments are relevant in the context of ice-structure interaction.

Page 2, line 26 – Why has it become increasingly important to use time-dependent constitutive modeling. When was it less important? Perhaps, the authors are referring to recent drastic changes in the Arctic sea ice. The sentence here is rather vague.

Page 2, line 30 – While it is true that ice sheet and glacier modelers use viscous creep law, the terms long term and short term are vaguely defined. As my research has found, sometime a few hours is all that takes for viscous behavior to dominate, which is not really that long term. Please explain clearly that short time scales you mean are seconds or minutes or hours

Page 3, lines 65 to 70 – The study's aims are noted here. However, there is an important discussion missing here about viscoelastic fracture mechanics. The concept of fracture toughness or critical stress intensity factor is only well defined for linear elastic solids or elasto-plastic solids with small scale yielding. The authors should state and explain the definitions. Of the apparent fracture toughness $K_Q$ and the loading rate $\dot{K}$, and why they are relevant quantify to ice mechanical behavior. What are the specific assumptions made about the ice viscoelastic behavior. Refer to any experiments and modeling studies in the literature that establish the theory of fracture in

time-dependent materials.

Creep-recovery fracture experiments

Page 3, lines 75 to 85 – The scale of these experiments is truly impressive, however, referring to my previous why is mode I fracture relevant for ice-structure interaction. Aren't sea ice floes breaking up due to compression and plate buckling processes. Please explain the motivation for these experiments and how it can be used in large-scale modeling of ice-floe structure interaction. For example, will this study provide necessary parameters for discrete element modeling of sea ice-structure or ice-ship interaction.

Page 3, line 85 – The top surface temperature is noted as -2 C, but in Figure 2a the temperature below the surface is around -0.3 C. I am confused, please explain.

Page 3, line 86 – Please provide some more description of the experimental setup, ice growth etc as we still do not have access to your paper in press. Why does the grain size increase with depth? Also, how realistic is this for sea ice as opposed to stagnate lake ice with no waves.

Page 3, line 90 – The experiments report the load values and peak loads. However, it would be useful from a modeling perspective to get crack initiation stress. Is it possible that this sort of information can be extracted and reported from experiments. This will make the paper's results useful to those modeling sea ice-structure interaction.

Page 4, line 95 – How do the applied load rates and load levels related to real ice floes. A bit more justification is needed to establish the rationale for testing.

Page 4, line 103 – I am failing to understand the purpose of creep loadings. If the creep loads were kept small so that no damage nucleates and with recovery periods, there should not affect. In fact, this is what is observed with the results.

Page 4, line 108 – Once again how do these cyclic load levels and loading rates related to the physical setting. Are these in any way representative of the ocean wave loads

on sea ice floes?

Nonlinear time-dependent modeling

Page 5, Equation 2 – Replacing the stress and strain with load and displacement is valid only for linear behavior. Has Schapery's model used with load and displacement before in any literature?

Page 7, line 195 – What is the purpose of the modeling and parameter estimation. I ask this because I work in ice fracture modeling and cannot really see how these experiments can improve the fracture models.

Results and discussion

Page 7, line 204 – How is the weight function approach applied? Numerical evaluation of integrals with weight function approach can lead to inconsistencies. Why not use the displacement correlation method directly using COD and CMOD and NCOD?

Page 8, line 213 – Figure 5a needs more explanation. In viscoelastic materials, the peak load increases with loading rate. Please define precisely what Kˆdot is and why the peak load decreases as you increase Kˆdot. Also, defined what you mean by failure load. Is it the same as peak load? If so, then just use one terminology consistently.

Page 8, line 228 – What are the differences in the post-peak load curves that should be identified. Is it the oscillatory nature of load displacement curves in cyclic sequences? A better explanation would be useful.

Page 8, line 235 – The authors state "It is clear from Figs. 7b and 8b . . ." How is it clear? The writing style is a bit confusing.

Page 9, line 243 – I only know of the Maxwell model and the generalized Maxwell model. What is a simple Maxwell model?

Page 9, lines 247 to 259 – This whole paragraph should be written as a separate discussion section. Based on my recollection the experiments of Sinha and Cole involved

compression loads and not tension loads, and there were not really on pre-cracks ice slabs. This lead to the question on why delayed elastic effect was not there? However, it is not clear why this is even an important question in the context of ice-structure interaction.

Page 267 – The statement "When the specimen dimensions are several meters, apparently viscoelasticity is not an important deformation component" is poorly explained. Also, what is the consequence of this finding? Is the author suggestion that one can just use elastic model for sea ice-interaction? Is there any relevance of these results for floating ice shelves, which are much larger than ice floes?

Conclusion

Overall, I am not clear on what the broader purpose of the paper is? Why did the author's select the specimen size and loading rates they used. Why specifically test creep/cyclic recovery? How is this work relevant to the motivation mentioned in the first paragraph of the introduction – interaction of ice floes with structure. How to use the data and findings of this paper in any future modeling analysis. A comprehensive revision of this article is needed and I recommend including a discussion section to address the implications of this research.

---

## Author Comment (AC1) · 30 Nov 2020

We thank the reviewer for carefully reviewing our work and making constructive comments. We appreciate all the time and efforts he/she put in their thorough review. All the reviewer's comments were considered in the revised manuscript. Detailed answers to each comment are given below.

1. RC1: The authors go further, however, weakening/endangering the manuscript. They analyze their data in terms of a constitutive model that was developed by Schapery (1969, 1997) for uniaxial loading (noted in lines 171-173). But in the experiments at hand, deformation occurred under a multiaxial stress state. Given that the model and the data relate to different states of stress and given that values of the

[Figure]

many (eight) unknown parameters in the model were derived by fitting the data and not from independent measurements, it is difficult to accept the statement (lines 303-305) that the analysis " provides a firm support of the ability of Schapery's constitutive model to describe the time-dependent response of columnar freshwater S2 ice up to crack growth initiation."

Authors: The Schapery model is developed for a uniaxial normal stress state. In our experimental set up (Fig. 1 in the manuscript), the response of the test specimen is dominated by the normal stresses at the direction normal to the X-axis, ahead of the crack. This stress state can be approximated as uniaxial in the same way as in beam bending; the stress is uniaxial tension at the crack tip and then changes linearly. Adamson and Dempsey [1] have successfully used the same type of modelling for a similar setup. In addition, Schapery's model has been verified and validated by LeClair [2]. Other analyses and approaches didn't do as well. For instance, the experimental data by LeClair [2] was modelled by Schapery's model [2] and by a physically-based FE model [3]. While the physical model did a reasonable job of modelling the data, Schapery's straight-forward model did a better job.

The approach we have used – fitting a model with experimental data by using optimization – is common in fracture models with several parameters. Pure experimental methods to determine these parameters have proven extremely difficult and indirect methods, based on parametric fitting, has been developed and used instead [4].

2. RC1: It is even more difficult to accept the claim (lines 242-243) that under the conditions of these experiments "there is no delayed elastic recovery".

Authors: We understand the reviewer's reluctance to accept the lack of delayed elastic recovery. What we measured for freshwater ice has not been reported earlier. However, this surprising response is what we measured in these tests (Figs 7b and 8b in the manuscript). Further studies are needed to confirm the result and to find explanations for the behaviour. It may be important that, compared with earlier work on freshwater

S2 ice, our samples were large and very warm.

3. RC1: The title presents a problem: it is misleading. This is not the kind of experiment that allows a characterization of elastic-viscoelastic deformation of ice. Rather, as already noted, it allows a conclusion to be made on fracture toughness and its insensitivity to pre-strain. The title needs to be changed to reflect that finding.

Authors: We will change the title into "Creep and fracture of warm columnar freshwater ice" and hope that it reflects the content better than the original title.

4. RC1: The other problem is that the manuscript contains a contradiction. The claim that the experiments were performed on ice at -2 C contradicts the temperature profile shown in Fig. 3a. There, where temperature is plotted versus depth (from 0 to 35 cm) in the ice, temperature ranges from -0.3 C near the top to 0 C near the bottom.

Authors: We apologize for the confusion. During the experiments, the ambient temperature in the laboratory was kept at -2 C. The temperature profile within the ice is shown in Fig. 2a in the manuscript. The text is edited to clear up this confusion.

5. RC1: Finally, it would be helpful to know in which journal the repeated reference to Gharamti et al. is "in press".

Authors: The paper by Gharamti et al. [5] is now published. The cited reference is edited.

References

1. Adamson RM, Dempsey JP. Field-scale in-situ compliance of arctic first-year sea ice. Journal of Cold Regions Engineering 1998;12:52–63.

2. LeClair ES, Schapery RA, Dempsey JP. A broad-spectrum constitutive modeling technique applied to saline ice. International Journal of Fracture 1999;97:209–26.

3. O'Connor D, West B, Haehnel R, Asenath-Smith E, Cole D. A viscoelastic integral formulation and numerical implementation of an isotropic constitutive model of saline

ice. Cold Regions Science and Technology 2020;171:102983.

4. Elices M, Guinea GV, Gomez J, Planas J. The cohesive zone model: advantages, limitations and challenges. Engineering Fracture Mechanics 2002;69:137–63.

5. Gharamti IE, Dempsey JP, Polojärvi A, Tuhkuri J. Fracture of S2 columnar freshwater ice: size and rate effects. Acta Materialia 2021;202:22–34.

---

## Author Comment (AC2) · 30 Nov 2020

We thank the reviewer for carefully reviewing our work and making constructive comments. We appreciate all the time and efforts he/she put in their thorough review. All the reviewer's comments were considered in the revised manuscript. Detailed answers to each comment are given below.

1. RC2: Page 1, line 14 – The eventual goal of these experiments seems to be to better describe or model ice loads during sea ice floe interactions with structures in the Arctic. As the authors mention the deformation modes are quite complex during ice-structure interaction, but there is no literature cited on this work. For example, Claude Daley at MUN has done experimental and modeling work on this using ice indentation experi-

[Figure]

ments on structures, which seems to be the most relevant mechanism for transfer of ice loads onto structures. One would rarely imagine a floating ice flow to be subjected to mode I tension. Please add more description to the introduction about what experimental data is available, what motivated the current experimental study and why mode I fracture experiments are relevant in the context of ice-structure interaction.

Authors: Many of the comments by the reviewer deal with sea ice, Arctic and engineering relevance. It appears that the introduction of our submission was unfortunately giving an impression that our work on creep deformation and fracture of freshwater ice has direct application in sea ice and Arctic engineering. That is not the case and we apologize for the confusion caused. Deformation and fracture of ice are highly dependent on salinity, temperature, strain rate, sample size, grain type, and grain size. Our paper reports results from laboratory experiments which were conducted to study the creep and fracture of warm, floating, columnar grained S2 freshwater ice. The work is directly relevant to a number of problems on freshwater ice in rivers and lakes [1] and the Baltic Sea which is almost freshwater ice. However, it has also general relevance to the creep and fracture of a quasi-brittle material. Unless we restrict our interest on the short time scales where only elastic response is relevant, the creep deformations must be modeled to obtain the true fracture behavior. In materials with time-dependent properties, the fracture and creep deformations are coupled.

Mode I loading is rather common in a number of ice problems. For example, sea ice floes fracture when in contact with ships and offshore structures or when loaded by waves, river ice fractures during interaction with bridge piers, and thermal cracks form in lakes and reservoirs.

2. RC2: Page 2, line 26 – Why has it become increasingly important to use time-dependent constitutive modeling. When was it less important? Perhaps, the authors are referring to recent drastic changes in the Arctic sea ice. The sentence here is rather vague.

Authors: We did think of the warming climate and thus warming ice which may poten-tially increase the importance of creep deformation and apologize for not writing this clearly. In addition, the applications like river ice breakup happen in late spring and the ice is very warm. For that reason, the creep deformations are very important but historically cold ice has been studied typically.

3. RC2: Page 2, line 30 – While it is true that ice sheet and glacier modelers use viscous creep law, the terms long term and short term are vaguely defined. As my research has found, sometime a few hours is all that takes for viscous behavior to dominate, which is not really that long term. Please explain clearly that short time scales you mean are seconds or minutes or hours.

Authors: We agree that "long" and "short" are vaguely defined terms and have different meaning in different contexts. It is also not correct to imply that this study is relevant to glaciers. We are studying columnar grained ice not very fine grained equiaxed snow ice.

4. RC2: Page 3, lines 65 to 70 – The study's aims are noted here. However, there is an important discussion missing here about viscoelastic fracture mechanics. The concept of fracture toughness or critical stress intensity factor is only well defined for linear elastic solids or elasto-plastic solids with small scale yielding. The authors should state and explain the definitions. Of the apparent fracture toughness $K_Q$ and the loading rate $\hat{K}$dot, and why they are relevant quantity to ice mechanical behavior. What are the specific assumptions made about the ice viscoelastic behavior. Refer to any experiments and modeling studies in the literature that establish the theory of fracture in time-dependent materials.

Authors: We will add a discussion on time-dependent fracture. The reviewer is correct. It has been known that the viscoelastic fracture mechanics [2] is on a firm foundation so long as a finite cohesive zone is attached to the traction-free crack tip. The one-parameter fracture mechanics encompassed by the $K_Q$ notation is not applicable [3]

TCD

and will be removed. Although Kˆdot was used for comparative measure of the loading rate, it will be replaced by the measured time of failure in each experiment. The K_Q plot will be replaced by a plot of the peak loads. We hope that any confusion will be cleared up then.

5. RC2: Page 3, lines 75 to 85 – The scale of these experiments is truly impressive, however, referring to my previous why is mode I fracture relevant for ice-structure interaction. Aren't sea ice floes breaking up due to compression and plate buckling processes. Please explain the motivation for these experiments and how it can be used in largescale modeling of ice-floe structure interaction. For example, will this study provide necessary parameters for discrete element modeling of sea ice-structure or ice-ship interaction.

Authors: Please see our response to Comment 1. Our main concern here is how tensile cracks develop in columnar freshwater ice under the applied loading.

6. RC2: Page 3, line 85 – The top surface temperature is noted as -2 C, but in Figure 2a the temperature below the surface is around -0.3 C. I am confused, please explain.

Authors: We apologize for the confusion. During the experiments, the ambient temperature in the laboratory was kept at -2 C. The temperature profile within the ice is shown in Fig. 2a in the manuscript. The text is edited to clear this confusion.

7. RC2: Page 3, line 86 – Please provide some more description of the experimental setup, ice growth etc as we still do not have access to your paper in press. Why does the grain size increase with depth? Also, how realistic is this for sea ice as opposed to stagnate lake ice with no waves.

Authors: The paper by Gharamti et al. [4] is now published. The cited reference is edited. The grain size increasing with depth is the characteristic of columnar S2 ice [5]. Michel and Ramseier [5] classified lake and river ice according to the size, shape and orientation of the crystals and the environmental factors causing them. For quiet lakes

subject to no wind and no snow falling, S1 ice with vertical c-axis orientation will form. The paper will be revised with a thorough discussion of the grain size effect.

8. RC2: Page 3, line 90 – The experiments report the load values and peak loads. However, it would be useful from a modeling perspective to get crack initiation stress. Is it possible that this sort of information can be extracted and reported from experiments. This will make the paper's results useful to those modeling sea ice-structure interaction.

Authors: We cannot determine the crack-initiation stress in these creep and cyclic-recovery experiments. The crack-initiation stress can be computed for the monotonic DC tests [4] because the stress-separation curve was derived for the DC tests. However, for the LC tests here, the stress-separation curve is unknown.

9. RC2: Page 4, line 95 – How do the applied load rates and load levels related to real ice floes. A bit more justification is needed to establish the rationale for testing.

Authors: We are mainly concerned here with the tensile cracks growing in columnar freshwater ice. We could be studying the bearing capacity of lake ice, the splitting of lake ice by an ice breaker, the breakup of river ice, and similar applications in the Baltic sea etc. Ice in nature is loaded through a wide range of time scales. The loading was chosen to reflect one time-dependent response that can be encountered. The loading rate used is similar than used in earlier sea ice studies and thus allows comparison of these two materials.

10. RC2: Page 4, line 103 – I am failing to understand the purpose of creep loadings. If the creep loads were kept small so that no damage nucleates and with recovery periods, there should not affect. In fact, this is what is observed with the results.

Authors: The creep-recovery tests in the time-domain were conducted to study the response of the ice under several load steps, which has not been studied before on freshwater ice. Previous studies have concentrated either on cyclic loading in the frequency domain [6,7] or on a single load step [8]. The main reason the loads were kept

small is to avoid damage [9] because we are not modelling damage. The creep/cyclic-recovery sequence did affect the accumulation of the viscoplastic component of the crack opening displacement (Fig. 11 in the manuscript).

11. RC2: Page 4, line 108 – Once again how do these cyclic load levels and loading rates related to the physical setting. Are these in any way representative of the ocean wave loads on sea ice floes?

Authors: Please see our response to Comment 9.

12. RC2: Page 5, Equation 2 – Replacing the stress and strain with load and displacement is valid only for linear behavior. Has Schapery's model used with load and displacement before in any literature?

Authors: The same modelling for load and displacement has been used by Adamson and Dempsey [10]. The modelling works very well, and the CMOD was predicted correctly until the crack began to propagate.

13. RC2: Page 7, line 195 – What is the purpose of the modeling and parameter estimation. I ask this because I work in ice fracture modeling and cannot really see how these experiments can improve the fracture models.

Authors: The approach we have used – fitting a model with experimental data by using optimization – is common in fracture models with several parameters. Pure experimental methods to determine these parameters have proven extremely difficult and indirect methods, based on parametric fitting, has been developed and used instead [11]. In addition, the Schapery model can fit successfully both creep-recovery and cyclic loading.

14. RC2: Page 7, line 204 – How is the weight function approach applied? Numerical evaluation of integrals with weight function approach can lead to inconsistencies. Why not use the displacement correlation method directly using COD and CMOD and NCOD?

Authors: A lot of work has been done on the weight function of an edge-cracked rectangular plate [12] used in the current experiments. The accuracy of this weight function was derived, assessed and thoroughly validated by comparison with other published data. The displacement correlation method cannot be used because of the presence of creep. The deformations are affected by creep and by possible growth of the crack.

15. RC2: Page 8, line 213 – Figure 5a needs more explanation. In viscoelastic materials, the peak load increases with loading rate. Please define precisely what $\hat{K}$dot is and why the peak load decreases as you increase $\hat{K}$dot. Also, defined what you mean by failure load. Is it the same as peak load? If so, then just use one terminology consistently.

Authors: In these experiments, the failure load is the same as the peak load. We are not including $K\_Q$ and $\hat{K}$dot in the analysis anymore.

16. RC2: Page 8, line 228 – What are the differences in the post-peak load curves that should be identified. Is it the oscillatory nature of load displacement curves in cyclic sequences? A better explanation would be useful.

Authors: The decay of the load for the creep-recovery tests (Fig. 5b in the manuscript) took a much longer time than that for the monotonic tests (Fig. 5a in the manuscript). Unfortunately, the reason for the oscillatory nature of the signal is unclear to us.

17. RC2: Page 8, line 235 – The authors state "It is clear from Figs. 7b and 8b : : :" How is it clear? The writing style is a bit confusing.

Authors: The measured displacement records in Figs. 7b and 8b show the absence of the viscoelastic component. A typical creep displacement-time record displaying the three displacement components (elastic, viscoelastic and viscous) looks as shown in Fig. 1 here. By simply comparing Fig. 1 here with each creep/cyclic-recovery sequence in Figs. 7b and 8b in the manuscript, one can deduce that the viscoelastic displacement and the viscoelastic recovery were absent from the current data.

[Figure]

18. RC2: Page 9, line 243 – I only know of the Maxwell model and the generalized Maxwell model. What is a simple Maxwell model?

Authors: We meant a Maxwell model composed of a nonlinear spring and a nonlinear dashpot. To avoid confusion, this statement is deleted in the revised text.

19. RC2: Page 9, lines 247 to 259 – This whole paragraph should be written as a separate discussion section. Based on my recollection the experiments of Sinha and Cole involved compression loads and not tension loads, and there were not really on pre-cracks ice slabs. This lead to the question on why delayed elastic effect was not there? However, it is not clear why this is even an important question in the context of ice-structure interaction.

Authors: The authors thank the reviewer for his/her recommendation. The authors created a separate discussion section.

20. RC2: Page 267 – The statement "When the specimen dimensions are several meters, apparently viscoelasticity is not an important deformation component" is poorly explained. Also, what is the consequence of this finding? Is the author suggestion that one can just use elastic model for sea ice-interaction? Is there any relevance of these results for floating ice shelves, which are much larger than ice floes?

Authors: Our experiments suggest that for the large sample size and the kind of ice studied (very warm freshwater ice) under the loading applied, the response was elastic-viscoplastic. More experiments are needed to make more general conclusions.

21. RC2: Overall, I am not clear on what the broader purpose of the paper is? Why did the author's select the specimen size and loading rates they used. Why specifically test creep/cyclic recovery? How is this work relevant to the motivation mentioned in the first paragraph of the introduction – interaction of ice floes with structure. How to use the data and findings of this paper in any future modeling analysis. A comprehensive revision of this article is needed and I recommend including a discussion section to

address the implications of this research.

Authors: Please see our response to Comment 1.

References

1. Ashton GD. River and lake ice engineering. Water Resources Publication, Littletown, Colorado; 1986.

2. Kostrov BV, Nikitin LV. Some general problems of mechanics of brittle fracture. Archiwum Mechaniki Stosowanej 1970;22, English Version:749–76.

3. Dempsey JP, Cole DM, Wang S. Tensile fracture of a single crack in first-year sea ice. Philosophical Transactions of the Royal Society A 2018;376(2129):20170346.

4. Gharamti IE, Dempsey JP, Polojärvi A, Tuhkuri J. Fracture of S2 columnar freshwater ice: size and rate effects. Acta Materialia 2021;202:22–34.

5. Michel B, Ramseier R. Classification of river and lake ice. Canadian Geotechnical Journal 1971;8(1):36–45.

6. Cole DM. Reversed direct-stress testing of ice: Initial experimental results and analysis. Cold Regions Science and Technology 1990;18:303–21.

7. Cole DM. A model for the anelastic straining of saline ice subjected to cyclic loading. Philosophical Magazine A 1995;72(1):231–48.

8. Sinha NK. Rheology of columnar-grained ice. Experimental Mechanics 1978;18:464–70.

9. Kachanov L. Introduction to continuum damage mechanics; vol. 10. Springer Science & Business Media; 1986.

10. Adamson RM, Dempsey JP. Field-scale in-situ compliance of arctic first-year sea ice. Journal of Cold Regions Engineering 1998;12:52–63.

11. Elices M, Guinea GV, Gomez J, Planas J. The cohesive zone model: advantages,

limitations and challenges. Engineering Fracture Mechanics 2002;69:137–63.

12. Dempsey JP, Mu Z. Weight function for an edge-cracked rectangular plate. Engineering Fracture Mechanics 2014;132:93–103.

**Fig. 1.** General displacement-time record for a creep-recovery test. The elastic, viscoelastic, and viscous (viscoplastic) components are marked.

---

## Author Response (AR1)

**Authors' response file**

I.E. Gharamti, J.P. Dempsey, A. Polojärvi and J. Tuhkuri

January 13, 2021
* * *
**Replies to reviewers' comments**

We thank the editor and the reviewers for carefully reviewing our work and making constructive comments. We appreciate all the time and efforts they put in their thorough review. All the reviewer comments were considered in the revised manuscript. Detailed answers to each comment are given below.

**1. General modifications in the revised manuscript**

- Added text is displayed in red.

- Deleted sentences are marked with a red strikethrough.

**2. Reviewer 1 comments**

1. The authors go further, however, weakening/endangering the manuscript. They analyze their data in terms of a constitutive model that was developed by Schapery (1969, 1997) for uniaxial loading (noted in lines 171-173). But in the experiments at hand, deformation occurred under a multiaxial stress state. Given that the model and the data relate to different states of stress and given that values of the many (eight) unknown parameters in the model were derived by fitting the data and not from independent measurements, it is difficult to accept the statement (lines 303-305) that the analysis " provides a firm support of the ability of Schapery's constitutive model to describe the time-dependent response of columnar freshwater S2 ice up to crack growth initiation."

The Schapery model is developed for a uniaxial normal stress state. In our experimental set up (Fig. 1), the response of the test specimen is dominated by the normal stresses at the direction normal to the X-axis, ahead of the crack. This stress state can be approximated as uniaxial in the same way as in beam bending; the stress is uniaxial tension at the crack tip and then changes linearly. Adamson and Dempsey [1] have succesfully used the same type of modelling for a similar setup. In addition, Schapery's model has been verfied and validated by LeClair [2]. Other analyses and approaches didn't do as well. For instance, the experimental data by LeClair [2] was modelled by Schapery's model [2] and by a physically-based FE model [3]. While the physical model did a reasonable job of modelling the data, Schapery's straight-forward model did a better job.

The approach we have used – fitting a model with experimental data by using optimization – is common in fracture models with several parameters. Pure experimental methods to determine these parameters have proven extremely difficult and indirect methods, based on parametric fitting, has been developed and used instead [4].

Some text is added in that context.

2. It is even more difficult to accept the claim (lines 242-243) that under the conditions of these experiments "there is no delayed elastic recovery".

We understand the reviewer's reluctance to accept the lack of major delayed elastic recovery. What we measured for freshwater ice has not been reported earlier. However, this surprising response of no significant viscoelasticity is what we measured in these tests (Figs 7b and 8b). Further studies are needed to confirm the result and to find explanations for the behaviour. It may be important that, compared with earlier work on freshwater S2 ice, our samples were large and very warm.

3. The title presents a problem: it is misleading. This is not the kind of experiment that allows a characterization of elastic-viscoelastic deformation of ice. Rather, as already noted, it allows a conclusion to be made on fracture toughness and its insensitivity to pre-strain. The title needs to be changed to reflect that finding.

We changed the title into "Creep and fracture of warm columnar freshwater ice" and hope that it reflects the content better than the original title.

4. The other problem is that the manuscript contains a contradiction. The claim that the experiments were performed on ice at -2°C contradicts the temperature profile shown in Fig. 3a. There, where temperature is plotted versus depth (from 0 to 35 cm) in the ice, temperature ranges from -0.3 °C near the top to 0°C near the bottom.

We apologize for the confusion. During the experiments, the ambient temperature in the laboratory was kept at -2°C. The temperature profile within the ice is shown in Fig. 2a. The text is edited to clear up this confusion.

5. Finally, it would be helpful to know in which journal the repeated reference to Gharamti et al. is "in press".

The paper by Gharamti et al. [5] is now published. The cited reference is edited.

**3. Reviewer 2 comments**

1. Page 1, line 14 – The eventual goal of these experiments seems to be to better describe or model ice loads during sea ice floe interactions with structures in the Arctic. As the authors mention the deformation modes are quite complex during ice-structure interaction, but there is no literature cited on this work. For example, Claude Daley at MUN has done experimental and modeling work on this using ice indentation experiments on structures, which seems to be the most relevant mechanism for transfer of ice loads onto structures. One would rarely imagine a floating ice flow to be subjected to mode I tension. Please add more description to the introduction about what experimental data is available, what motivated the current experimental study and why mode I fracture experiments are relevant in the context of ice-structure interaction.

Many of the comments by the reviewer deal with sea ice, Arctic and engineering relevance. It appears that the introduction of our submission was unfortunately giving an impression that our work on creep deformation and fracture of freshwater ice has direct application in sea ice and Arctic engineering. That is not the case and we

apologize for the confusion caused. Deformation and fracture of ice are highly dependent on salinity, temperature, strain rate, sample size, grain type, and grain size. Our paper reports results from laboratory experiments which were conducted to study the creep and fracture of warm, floating, columnar grained S2 freshwater ice. The work is directly relevant to a number of practical problems on columnar freshwater ice [6]. However, it has also general relevance to the creep and fracture of a quasi-brittle material. Unless we restrict our interest on the short time scales where only elastic response is relevant, the creep deformations must be modeled to obtain the true fracture behavior. In materials with time-dependent properties, the fracture and creep deformations are coupled.

Mode I loading is rather common in a number of ice problems. For example, freshwater ice sheets fracture when in contact with ships, river ice fractures during interaction with bridge piers, and thermal cracks form in lakes and reservoirs.

2. Page 2, line 26 – Why has it become increasingly important to use time-dependent constitutive modeling. When was it less important? Perhaps, the authors are referring to recent drastic changes in the Arctic sea ice. The sentence here is rather vague.

We did think of the warming climate and thus warming ice which may potentially increase the importance of creep deformation and apologize for not writing this clearly. In addition, the applications like river ice breakup happen in late spring and the ice is very warm. For that reason, the creep deformations are very important but historically cold ice has been studied typically.

3. Page 2, line 30 – While it is true that ice sheet and glacier modelers use viscous creep law, the terms long term and short term are vaguely defined. As my research has found, sometime a few hours is all that takes for viscous behavior to dominate, which is not really that long term. Please explain clearly that short time scales you mean are seconds or minutes or hours.

We agree that "long" and "short" are vaguely defined terms and have different meaning in different contexts. The text is edited. It is also not correct to imply that this study is relevant to glaciers. We are studying columnar grained ice not very fine grained equiaxed snow ice.

4. Page 3, lines 65 to 70 – The study's aims are noted here. However, there is an important discussion missing here about viscoelastic fracture mechanics. The concept of fracture toughness or critical stress intensity factor is only well defined for linear elastic solids or elasto-plastic solids with small scale yielding. The authors should state and explain the definitions. Of the apparent fracture toughness $K_Q$ and the loading rate $\dot{K}$, and why they are relevant quantity to ice mechanical behavior. What are the specific assumptions made about the ice viscoelastic behavior. Refer to any experiments and modeling studies in the literature that establish the theory of fracture in time-dependent materials.

We added a discussion on time-dependent fracture. The reviewer is correct. It has been known that the viscoelastic fracture mechanics [7] is on a firm foundation so long as a finite cohesive zone is attached to the traction-free crack tip. The one-parameter fracture mechanics encompassed by the $K_Q$ notation is not applicable [8] and is removed. Although $\dot{K}$ was used for comparative measure of the loading rate, it is replaced by the measured time of failure in each

experiment. The $K_Q$ plot is replaced by a plot of the peak loads. We hope that any confusion will be cleared up then.

5. Page 3, lines 75 to 85 – The scale of these experiments is truly impressive, however, referring to my previous why is mode I fracture relevant for ice-structure interaction. Aren't sea ice floes breaking up due to compression and plate buckling processes. Please explain the motivation for these experiments and how it can be used in largescale modeling of ice-floe structure interaction. For example, will this study provide necessary parameters for discrete element modeling of sea ice-structure or ice-ship interaction.

Please see our response to Comment 1. Our main concern here is how tensile cracks develop in columnar freshwater ice under the applied loading.

6. Page 3, line 85 – The top surface temperature is noted as -2 C, but in Figure 2a the temperature below the surface is around -0.3 C. I am confused, please explain.

We apologize for the confusion. During the experiments, the ambient temperature in the laboratory was kept at -2°C. The temperature profile within the ice is shown in Fig. 2a. The text is edited to clear this confusion.

7. Page 3, line 86 – Please provide some more description of the experimental setup, ice growth etc as we still do not have access to your paper in press. Why does the grain size increase with depth? Also, how realistic is this for sea ice as opposed to stagnate lake ice with no waves.

The paper by Gharamti et al. [5] is now published. The cited reference is edited.

The grain size increasing with depth is the characteristic of columnar ice [9]. Cole [10] wrote: "In columnar ice, the average grain diameter in the horizontal plane typically increases with depth because the faster growing, c-axis horizontal grains systematically eliminate the grains least favorably oriented for growth."

Michel and Ramseier [9] classified lake and river ice according to the size, shape and orientation of the crystals and the environmental factors causing them. For quiet lakes subject to no wind and no snow falling, S1 ice with vertical c-axis orientation will form. The paper will be revised with a thorough discussion of the microstucture (grain size effect).

8. Page 3, line 90 – The experiments report the load values and peak loads. However, it would be useful from a modeling perspective to get crack initiation stress. Is it possible that this sort of information can be extracted and reported from experiments. This will make the paper's results useful to those modeling sea ice-structure interaction.

We cannot determine the crack-initiation stress in these creep and cyclic-recovery experiments. The crack-initiation stress can be computed for the monotonic DC tests [5] because the stress-separation curve was derived for the DC tests. However, for the LC tests here, the stress-separation curve is unknown.

9. Page 4, line 95 – How do the applied load rates and load levels related to real ice floes. A bit more justification is needed to establish the rationale for testing.

We are mainly concerned here with the tensile cracks growing in columnar freshwater ice. We could be studying the bearing capacity of lake ice, the splitting of lake ice by an ice breaker, the breakup of river ice, etc. Ice in nature is

loaded through a wide range of time scales. The loading was chosen to reflect one time-dependent response that can be encountered. The loading rate used is similar than used in earlier sea ice studies and thus allows comparison of these two materials.

10. Page 4, line 103 – I am failing to understand the purpose of creep loadings. If the creep loads were kept small so that no damage nucleates and with recovery periods, there should not affect. In fact, this is what is observed with the results.

The creep-recovery tests in the time-domain were conducted to study the response of the ice under several load steps, which has not been studied before on freshwater ice. Previous studies have concentrated either on cyclic loading in the frequency domain [11, 12] or on a single load step [13].

The main reason the loads were kept small is to avoid damage because we are not modelling damage [14]. The creep/cyclic-recovery sequence did affect the accumulation of the viscoplastic component of the crack opening displacement (Fig. 11).

11. Page 4, line 108 – Once again how do these cyclic load levels and loading rates related to the physical setting. Are these in any way representative of the ocean wave loads on sea ice floes?

Please see our response to Comment 9.

12. Page 5, Equation 2 – Replacing the stress and strain with load and displacement is valid only for linear behavior. Has Schapery's model used with load and displacement before in any literature?

The same modelling for load and displacement has been used by Adamson and Dempsey [1]. The modelling works very well, and the CMOD was predicted correctly until the crack began to propagate.

13. Page 7, line 195 – What is the purpose of the modeling and parameter estimation. I ask this because I work in ice fracture modeling and cannot really see how these experiments can improve the fracture models.

The approach we have used – fitting a model with experimental data by using optimization – is common in fracture models with several parameters. Pure experimental methods to determine these parameters have proven extremely difficult and indirect methods, based on parametric fitting, has been developed and used instead [4]. In addition, the Schapery model can fit successfully both creep-recovery and cyclic loading.

14. Page 7, line 204 – How is the weight function approach applied? Numerical evaluation of integrals with weight function approach can lead to inconsistencies. Why not use the displacement correlation method directly using COD and CMOD and NCOD?

A lot of work has been done on the weight function of an edge-cracked rectangular plate [15] used in the current experiments. The accuracy of this weight function was derived, assessed and thoroughly validated by comparison with other published data. The displacement correlation method cannot be used because of the presence of creep. The deformations are affected by creep and by possible growth of the crack.

15. Page 8, line 213 – Figure 5a needs more explanation. In viscoelastic materials, the peak load increases with loading rate. Please define precisely what K^dot is and why the peak load decreases as you increase K^dot. Also, defined what you mean by failure load. Is it the same as peak load? If so, then just use one terminology consistently. In these experiments, the failure load is the same as the peak load. We are not including $K$ and $\dot{K}$ in the analysis anymore.

16. Page 8, line 228 – What are the differences in the post-peak load curves that should be identified. Is it the oscillatory nature of load displacement curves in cyclic sequences? A better explanation would be useful.

The decay of the load for the creep-recovery tests (Fig. 5b) took a much longer time than that for the monotonic tests (Fig. 5a). Unfortunately, the reason for the oscillatory nature of the signal is unclear to us.

17. Page 8, line 235 – The authors state "It is clear from Figs. 7b and 8b : : :" How is it clear? The writing style is a bit confusing.

The measured displacement records in Figs. 7b and 8b show no significant viscoelastic component. A typical creep displacement-time record displaying the three displacement components (elastic, viscoelastic and viscous) looks as shown in Fig. 1. By simply comparing Fig. 1 here with each creep/cyclic-recovery sequence in Figs. 7b and 8b in the manuscript, one can deduce that the viscoelastic displacement and the viscoelastic recovery were not significant.

[Figure]

Figure 1: General displacement-time record for a creep-recovery test. The elastic, viscoelastic, and viscous (viscoplastic) components are marked.

18. Page 9, line 243 – I only know of the Maxwell model and the generalized Maxwell model. What is a simple Maxwell model?

We meant a Maxwell model composed of a nonlinear spring and a nonlinear dashpot. The word "simple" is deleted.

19. Page 9, lines 247 to 259 – This whole paragraph should be written as a separate discussion section. Based on my recollection the experiments of Sinha and Cole involved compression loads and not tension loads, and there were not really on pre-cracks ice slabs. This lead to the question on why delayed elastic effect was not there? However, it is not clear why this is even an important question in the context of ice-structure interaction.

The authors thank the reviewer for his/her recommendation. The authors created a separate discussion section.

20. Page 267 – The statement "When the specimen dimensions are several meters, apparently viscoelasticity is not an important deformation component" is poorly explained. Also, what is the consequence of this finding? Is the author suggestion that one can just use elastic model for sea ice-interaction? Is there any relevance of these results for floating ice shelves, which are much larger than ice floes?

Our experiments suggest that for the large sample size and the kind of ice studied (very warm freshwater ice) under the loading applied, the response was elastic-viscoplastic. More experiments are needed to make more general conclusions.

21. Overall, I am not clear on what the broader purpose of the paper is? Why did the author's select the specimen size and loading rates they used. Why specifically test creep/cyclic recovery? How is this work relevant to the motivation mentioned in the first paragraph of the introduction – interaction of ice floes with structure. How to use the data and findings of this paper in any future modeling analysis. A comprehensive revision of this article is needed and I recommend including a discussion section to address the implications of this research.

Please see our response to Comment 1.

[revised manuscript text omitted]

---

## Referee Report (RR1)

The purpose of the work described in the revised manuscript was to assess the time-dependent behavior of freshwater ice, including effects of creep and cyclic loading on fracture properties. To that end a joint experimental-modelling approach was adopted. Experiments were performed by prying open the sides of through-thickness notches that had been cut into large (3 x 6 meter), floating plates of warm (-0.3 C), S2 freshwater ice and then by measuring notch width versus time—the method the authors used (Gharamti et al. 2021) to explore whether size and/or loading rate affect the fracture toughness of the same material. The modelling part of the work was based on Schapery's constitutive theory for creep of polymeric material loaded uniaxially. Two findings: (i) creep and cycling sequences had no clear effect on failure load; and (ii) unlike in all past work where both recoverable (anelastic/delayed elastic/viscoelastic strain) and non-recoverable (viscoplastic) deformation have been found to contribute to creep of ice, significant viscoelasticity was not detected.

Finding (i) , with one proviso, is worth publishing. Earlier work (Rist et al. (1996, *Annals of Glaciology,* vol. 23, p.284), not referenced in this manuscript, hinted at an effect; now there appears not to be one. The proviso, in keeping with the stated objective of assessing fracture properties, is that this finding should be presented in terms of a fracture property, namely fracture toughness. Failure load is insufficient. The finding was presented in that way in the original manuscript (original Fig4). Why the change?

Finding (ii) is troubling. Prior work , correctly cited, revealed that viscoelasticity contributes to the creep of ice. Now, no significant anelasticity is detected. Why not? The authors point to differences in specimen size, temperature and grain size, although how those factors could account for the difference is not made clear. Instead, could the explanation reside in the experiment and analysis itself? Unravelling the various contributions to time-dependent deformation, even when experiments are made under uniaxial states of stress, is not easy. Here, the stress state is only approximately uniaxial and the unravelling is performed through mathematical manipulation to optimally fit model to data—a procedure the authors justify with the statement (lines 160-2): "This approach of fitting a model with experimental data is common in fracture models with several parameters. Pure experimental methods to establish these parameters has proven extremely difficult….". To conclude (lines 344-5) from this rather tortuous approach, justified by what could be read as an excuse, that the response of the ice was overall elastic-viscoplastic is questionable. The conclusion is not credible. Worse, it muddies the picture of the time-dependent deformation of ice.

---

## Author Response (AR2)

**Authors' response file**

I.E. Gharamti, J.P. Dempsey, A. Polojärvi and J. Tuhkuri

April 13, 2021
* * *
**Replies to reviewers' comments**

We thank the editor and the reviewers for carefully reviewing our work and making constructive comments. We appreciate all the time and efforts they put in their thorough review. All the reviewer comments were considered in the revised manuscript. Detailed answers to each comment are given below.

**1. General modifications in the revised manuscript**

- Added text is displayed in red.

- Deleted sentences are marked with a red strikethrough.

- Fig. 7d is added.

- New references are added: Dash et al. (2006), Gasdaska (1994), Muto and Sakai (1998) and Rist et al. (1996).

**2. Reviewer 1 comments**

Two findings: (i) creep and cycling sequences had no clear effect on failure load; and (ii) unlike in all past work where both recoverable (anelastic/delayed elastic/viscoelastic strain) and non-recoverable (viscoplastic) deformation have been found to contribute to creep of ice, significant viscoelasticity was not detected."

1. Finding (i) , with one proviso, is worth publishing. Earlier work (Rist et al. (1996, Annals of Glaciology, vol. 23, p.284), not referenced in this manuscript, hinted at an effect; now there appears not to be one. The proviso, in keeping with the stated objective of assessing fracture properties, is that this finding should be presented in terms of a fracture property, namely fracture toughness. Failure load is insufficient. The finding was presented in that way in the original manuscript (original Fig4). Why the change?

We thank the reviewer for his comment. However, the authors edited the revised manuscript and deleted the fracture toughness data in accordance with the comments given by the second reviewer of the first review round. The concept of fracture toughness or critical stress intensity factor is only well defined for linear elastic cases. The one-parameter fracture mechanics is not applicable in the context of time-dependent fracture mechanics, which is the current study case.

The authors replaced the fracture toughness plot by a plot of the peak/failure loads. The fracture toughness is function of the peak/failure load and geometry. As the tested specimens are of the same geometry, size, and crack length, the failure load gives a good indication about the effect of creep/cyclic loading on the fracture behavior.

We thank the reviewer for mentioning a paper by Rist et al. The authors checked the paper's relevance to the current study and decided to cite it in the current manuscript.

2. Finding (ii) is troubling. Prior work , correctly cited, revealed that viscoelasticity contributes to the creep of ice. Now, no significant anelasticity is detected. Why not? The authors point to differences in specimen size, temperature and grain size, although how those factors could account for the difference is not made clear. Instead, could the explanation reside in the experiment and analysis itself? Unravelling the various contributions to time-dependent deformation, even when experiments are made under uniaxial states of stress, is not easy. Here, the stress state is only approximately uniaxial and the unravelling is performed through mathematical manipulation to optimally fit model to data—a procedure the authors justify with the statement (lines 160-2): "This approach of fitting a model with experimental data is common in fracture models with several parameters. Pure experimental methods to establish these parameters has proven extremely difficult....". To conclude (lines 344-5) from this rather tortuous approach, justified by what could be read as an excuse, that the response of the ice was overall elastic-viscoplastic is questionable. The conclusion is not credible. Worse, it muddies the picture of the time-dependent deformation of ice.

The conclusion of no significant viscoelasticity is **MERELY** based on the experimental observations and measurements and **NOT** on the numerical modelling. The measured displacement-time records (Figs. 7b and 8b in the manuscript and Fig. R1a here) displayed clearly a constant creep rate that usually dominates in the secondary (steady-state) creep stage. This indicates an instantaneous transformation from the primary (transient) stage to the steady-state regime, which resulted in permanent (unrecoverable) displacement. Accordingly, the viscoelastic component which should develop in the primary creep stage was insignificant. The measured records resemble the response of a Maxwell model, consisting of a nonlinear spring and nonlinear dashpot, to a constant load step (Fig. R1b). The accumulation of the viscoplastic component and the insignificant viscoelasticity can be also seen from the hysteresis loops (Fig. 6 in the manuscript).

The results of the initial optimization trials supported the experimental observations. The viscoelastic component $\delta_{\text{CMOD}}^{ve}$ had no effect on the final fit between the data and the model (Fig. R2). The optimization algorithm fine-tuned $\kappa$ (see Eq. 11 in the manuscript) to a very small number ($10^{-18}$), indicating that the best model-data fit is attained when the viscoelastic term goes to zero. The final optimization runs were then carried out without considering the viscoelastic term in the numerical fitting.

This is a novel result for any type of ice. In comparison with earlier freshwater ice studies, most of the previous work (cited in the manuscript) used colder freshwater ice and smaller samples and reported significant viscoelasticity. In comparison with sea ice, the measured response of the current ice is different from what has been reported for sea ice. For example, Adamson and Dempsey [1] studied warm and floating sea ice, used similar size and testing setup as the current study, and reported a different behavior (Figs. R1c and R1d); the measured data showed a significant viscoelastic component by the decreasing strain rate.

The authors respect the reviewer's reluctance to accept the conclusion of no significant viscoelasticity. This is understandable as the conducted tests are new and unique and no similar results have been reported in the literature. The authors discussed (Section 5 in the manuscript) the possible factors that could have contributed to the observed behavior. More details are added in the revised manuscript. Viscoelasticity occurs normally when the internal stresses developing during loading at the local stress concentrations (triple points and grain boundary ledges) accommodate the grain boundary sliding and causes sliding in the reverse direction, giving rise to the recoverable component after unloading. However, in our case, the measurements showed that the grain boundary sliding produced permanent deformation. Several reasons can be pointed out, related to the temperature, microstructure, and nonlinear mechanisms in the process zone. Most importantly, our experiments used warmer ice and larger sizes than previous freshwater studies.

Concerning the effect of temperature: the warmer the temperature, the more liquid on the grain boundary. The high homologous test temperature (top ice surface ≈ -0.3 C) causes liquidity on the gain boundaries [2]. The intergranular melt phase on the grain boundary renders ice as two-phase polycrystal and significantly influences the creep and recovery process. In fact, the grain boundary sliding then consists of two processes: 1) the sliding of grains over one another and 2) the squeezing-in/out of the liquid between adjacent grain [3]. The shear behavior of the liquid film is function of its properties (thickness and amount). The presence of this liquid at the triple points and the boundary acted as a resisting obstacle for the grains to shear and deform back to their original form, resulting in the viscoplastic deformation.

The microstructure (grain size, crystalline texture) could be another contributing factor. Sinha [4] developed a nonlinear viscoelastic model, incorporating the grain size effect, to describe the high-temperature creep of polycrystalline materials. He concluded that delayed elastic strain exhibits an inverse proportionality with grain size. This suggests that the grain size (3-10 mm, Fig. 2b in the manuscript) of the ice samples is coarse enough not to produce any measurable viscoelastic deformation under the testing conditions. It is also probable that for this grain size, there was not enough local concentration points to arrest the grain boundary sliding and drive the recoverable and reverse sliding. In addition, [5] discussed that regularly ordered and packed microstructures limit the amount of sliding and rearrangement and lead to less anelastic strain. The ice growth in the Aalto Ice tank was very controlled and resulted in homogeneous ice sheet.

Finally, the nonlinear mechanisms in the process zone possibly relieved the internal stresses that are needed to accommodate the grain boundary sliding and drive the recovery. This implies that any microstructural damage that occurred during loading manifested as permanent deformation at the end of the test.

Concerning the specimen size, our experiments used larger sizes than previous freshwater studies. Testing the effect of size requires a test program that use different specimen sizes while keeping all the other conditions fixed. All the above-mentioned factors may have contributed to the measured response. However, the question as to which factor influenced mostly the measured behavior is an important research question and requires more experiments.

**3. Reviewer 2 comments**

1. While I recognize the novelty of the experimental studies and the usefulness of the collected data to the

[Figure]

Figure R1: (a) Measured displacement-time record of RP16. (b) Typical response of a Maxwell model, consisting of a nonlinear spring and nonlinear dashpot, to a constant load step.(c) Creep-recovery loading profile and the (d) corresponding crack-mouth-opening displacement measured record for a similar testing setup of in-situ first year sea ice [1].

[Figure]

Figure R2: CMOD-time record: Schapery's model with and without viscoelasticity versus the experimental data.

Cryosphere community, I still feel that the authors did not go far enough in terms of explaining how this data or the Schapery model can be used. Please add specific discussion about how this experimental data can be used by modelers and/or any planned future work.

The main concern from the conducted experiments is to gain a better understanding of the creep and fracture behavior of warm, foating and columnar grained freshwater ice. With the warming climate and thus warming ice, we are interested in creep deformations and how cracks develop in columnar freshwater ice.

Schapery's model has been singularly powerful and successful for this type of loading [1, 6]. Other analyses and approaches didn't do as well. For instance, the experimental data by LeClair [6] was modelled by Schapery's model [6] and by a physically-based FE model [7]. While the physical model did a reasonable job of modelling the data, Schapery's straight-forward model did a better job. However, before a general well-calibrated form of Schapery's model can be presented for columnar freshwater ice and used by future modelers, a lot of experimental work needs to be conducted at different testing conditions: temperatures, loading rates, sample sizes, etc.

2. The major finding is that viscoelasticity of S2 columnar ice is not significant because there is no delayed elastic recovery observed in the experiments (Figs. 7b and 8b). Therefore, the authors conclude that ice samples showed elastic-viscoplastic material behavior. The other reviewer seemed to be critical about this. The authors explanation "Further studies are needed to confirm this result and to find explanations for the behavior" is honestly a bit underwhelming. Could the authors instead explain what sort of further studies are necessary to confirm and find explanations. Can the authors test smaller colder samples and do a quick check? Considering that is a significant conclusion, it is essential that the authors unequivocally establish this.

The conducted large-scale tests are unique; no similar tests for columnar freshwater ice have been performed earlier. The obtained ice response is different from what has been reported earlier for freshwater ice. It is a novel result for any type of ice. In comparison with earlier freshwater ice studies, most of the previous work (cited in the manuscript) used colder freshwater ice and smaller samples and reported significant viscoelasticity. In comparison with sea ice, the measured response of the current ice is different from what has been reported for sea ice. For example, Adamson and Dempsey [1] studied warm and floating sea ice, used similar size and testing setup as the current study, and reported a different behavior (Figs. R1c and R1d); the measured data showed a significant viscoelastic component by the decreasing strain rate.

The authors discussed (Section 5 in the manuscript) the possible factors that could have contributed to the observed behavior (no significant viscoelasticity). More details are added in the revised manuscript. Viscoelasticity occurs normally when the internal stresses developing during loading at the local stress concentrations (triple points and grain boundary ledges) accommodate the grain boundary sliding and causes grain boundary sliding in the reverse direction, giving rise to the recoverable component after unloading. However, in our case, the measurements showed that the grain boundary sliding produced permanent deformation. Several reasons can be pointed out, related to the temperature, microstructure, and nonlinear mechanisms in the process zone. It is important to emphasize that in comparison with other tests, our ice specimens were large and very warm.

Concerning the effect of temperature: the warmer the temperature, the more liquid on the grain boundary. The high homologous test temperature (top ice surface $\approx$ -0.3 C) causes liquidity on the gain boundaries [2]. The

intergranular melt phase on the grain boundary renders ice as two-phase polycrystal and significantly influences the creep and recovery process. In fact, the grain boundary sliding then consists of two processes: 1) the sliding of grains over one another and 2) the squeezing-in/out of the liquid between adjacent grain [3]. The shear behavior of the liquid film is function of its properties (thickness and amount). The presence of this liquid at the triple points and the boundary can act as a resisting obstacle for the grains to shear and deform back to their original form.

The microstructure (grain size, crystalline texture) could be another contributing factor. [4] developed a nonlinear viscoelastic model, incorporating the grain size effect, to describe the high-temperature creep of polycrystalline materials. He concluded that delayed elastic strain exhibits an inverse proportionality with grain size. This suggests that the grain size (3-10 mm, Fig. 2b in the manuscript) of the ice samples is coarse enough not to produce any measurable viscoelastic deformation under the testing conditions. It is also probable that for this grain size, there was not enough local concentration points to arrest the grain boundary sliding and drive the recoverable and reverse sliding. In addition, [5] discussed that regularly ordered and packed microstructures limit the amount of sliding and rearrangement and lead to less anelastic strain. The ice growth in the Aalto Ice tank was very controlled and resulted in homogeneous ice sheet.

Finally, the nonlinear mechanisms in the process zone possibly relieved the internal stresses that are needed to accommodate the grain boundary sliding and drive the recovery. This implies that any microstructural damage that occurred during loading manifested as permanent deformation at the end of the test.

Concerning the specimen size, our experiments used larger sizes than previous freshwater studies. Testing the effect of size requires a test program that use different specimen sizes while keeping all the other conditions fixed. All the above-mentioned factors may have contributed to the measured response. However, the question as to which factor influenced mostly the measured behavior is an another research question and requires more experiments. Testing the effect of each factor requires a test program that considers this factor while keeping all the other conditions fixed.

We thank the reviewer for his suggestion of testing smaller and colder samples to do a quick check. However, the authors believe that a quick check is not possible; the current tests were conducted under very controlled growth and testing conditions in the Aalto Ice tank. In addition, testing of smaller and colder samples cannot directly help the analysis in the current study; especially that several earlier studies used smaller and colder samples and reported significant viscoelasticity (cited in the manuscript).

3. Page 1, line 15 – I appreciate the authors for clarify that they interest is freshwater ice sheets and not sea ice floes or Antarctic/Greenland glaciers or ice shelves. While the motivation for the experimental study is clear now, please clarify what the motivation is for calibrating the Schapery model instead of a cohesive zone model.

Schapery's model was used because the model has been singularly powerful and successful for this type of creep/cyclic-recovery loading [1, 6]. Other analyses and approaches didn't do as well in modelling the load and unload periods. For instance, the experimental data by LeClair [6] was modelled by Schapery's model [6] and by a physically-based FE model [7]. While the physical model did a reasonable job of modelling the data, Schapery's straight-forward model did a better job.

In a recent publication by the same authors [8], a viscoelastic cohesive zone model was applied to model similar experiments of the same ice under monotonic loading. Wang et al. doubted the ability of the cohesive zone model

to handle unloading [9]. They hinted at odd modelling results during the unloading phases. The back-calculated methodology of the cohesive model [10] is reliant on the monotonic growth of the process zone. This is problematic in the unloading phases where the cohesive model predicts a continuous increase in the process zone. In applying the cohesive zone model, the authors in [9] replaced the unloading phases by constant loading phases; assuming that the process zone stays constant during unloading.

In that sense, the ability of Schapery's model takes us a step further and allows the modelling of loading and unloading.

4. Page 6, line 189 – The authors referenced the work of Elices et al. (2002) that is entitled "Cohesive zone model: advantages, limitations, challenges." I am not sure how this paper is an appropriate citation for their argument about using indirect fitting methods. Please clarify and if you are using a specific argument/conclusion from Elices et al. (2002) put it in quotes.

Citing Elices et al. was added in the revised manuscript as a reply to reviewer 2 of the first review round. Elices et al. (2002) reviewed the cohesive zone model and included a discussion of the difficulties associated with direct experimental methods and the usage of indirect fitting methods instead. We cited their paper for the sole purpose of referring the reader to their discussion of parametric fitting. However, to avoid any confusion, the authors deleted the corresponding text from the revised manuscript.

5. Page 7, line 220 – Equation (11) is the main equation that relates CMOD with the applied load P. While the model agrees well with experimental data, wouldn't it be a better idea to calibrate a cohesive zone model and calibrate its parameters, so that it can be used for predicting ice fracture under realistic cases such as those mentioned in the first paragraph of the introduction..

See our answer to comment 3.

**References**

1. Adamson RM, Dempsey JP. Field-scale in-situ compliance of arctic first-year sea ice. *Journal of Cold Regions Engineering* 1998;12:52–63.

2. Dash J, Rempel A, Wettlaufer J. The physics of premelted ice and its geophysical consequences. *Reviews of modern physics* 2006;78(3):695.

3. Muto H, Sakai M. Grain-boundary sliding and grain interlocking in the creep deformation of two-phase ceramics. *Journal of the American Ceramic Society* 1998;81(6):1611–21.

4. Sinha NK. Grain boundary sliding in polycrystalline materials. *Philosophical Magazine A* 1979;40(6):825–42.

5. Gasdaska CJ. Tensile creep in an in situ reinforced silicon nitride. *Journal of the American Ceramic Society* 1994;77(9):2408–18.

6. LeClair ES, Schapery RA, Dempsey JP. A broad-spectrum constitutive modeling technique applied to saline ice. *International Journal of Fracture* 1999;97:209–26.

7. O'Connor D, West B, Haehnel R, Asenath-Smith E, Cole D. A viscoelastic integral formulation and numerical implementation of an isotropic constitutive model of saline ice. *Cold Regions Science and Technology* 2020;171:102983.

8. Gharamti IE, Dempsey JP, Polojärvi A, Tuhkuri J. Fracture of S2 columnar freshwater ice: size and rate effects. *Acta Materialia* 2021;202:22–34.

9. Wang S, Dempsey JP, Cole DM. In-situ fracture of first-year sea ice in mcmurdo sound: Test a2-sp8. In: *Proc. 18th Int. POAC Conf*; vol. 3. 2006:1071–82.

10. Mulmule S, Dempsey JP. A viscoelastic fictitious crack model for the fracture of sea ice. *Mechanics of Time-Dependent Materials* 1997;1(4):331–56.